# Multi-proxy reconstructions of May–September precipitation field in China over the past 500 years

Feng Shi[1,2*], Sen Zhao[3,4], Zhengtang Guo[1,5,6], Hugues Goosse[2], Qiuzhen Yin[2]

[1]Key Laboratory of Cenozoic Geology and Environment, Institute of Geology and Geophysics, Chinese Academy of Sciences, Beijing, 100029, China
[2]Georges Lemaître Centre for Earth and Climate Research, Earth and Life Institute, Université catholique de Louvain, Louvain-la-Neuve, 1348, Belgium
[3]Key Laboratory of Meteorological Disaster of Ministry of Education, and College of Atmospheric Science, Nanjing University of Information Science and Technology, Nanjing, 210044, China
[4]School of Ocean and Earth Sciences and Technology, University of Hawaii at Mānoa, Honolulu, HI, 96822, USA
[5]CAS Center for Excellence in Tibetan Plateau Earth Sciences, Beijing, 100101, China
[6]University of Chinese Academy of Sciences, Beijing, 100049, China

*Correspondence to*: Feng SHI (shifeng@mail.iggcas.ac.cn)

**Abstract.** The dominant modes of variability of precipitation for the whole China over the past millennium and the mechanism governing their spatial structure remain unclear. The first reason is probably that it is difficult to reconstruct the precipitation field in western China because the published high-resolution proxy records for this region are scarce. Numerous tree-ring chronologies have recently been archived in publicly available databases through PAGES2k activities, and these provide an opportunity to refine precipitation field reconstructions for China. Based on 479 proxy records, including 371 tree-ring width chronologies, a tree-ring isotope chronology, and 107 drought/flood indices, we reconstruct the precipitation field for China for the past half millennium using the optimal information extraction method. A total of 3631 of 4189 grid points in the reconstruction field passed the cross-validation process, accounting for 86.68% of the total number of grid points. The first leading mode of variability of the reconstruction shows coherent variations over most of China. The second mode, a north–south dipole in eastern China with variations of the same sign in western China and northern China, except for Xinjiang province, may be controlled by the El Niño-Southern Oscillation (ENSO) variability. The third mode, a "sandwich" triple mode in eastern China with variations of the same sign in western China and central China. Five of the six coupled ocean-atmosphere climate models (BCC-CSM1.1, CCSM4, FGOALS-s2, GISS-E2-R and MPI-ESM-P) of the Paleoclimate Modeling Intercomparison Project Phase III (PMIP3), can reproduce the south-north dipole mode of precipitation in eastern China, and its likely link with ENSO. However, there is mismatch in terms of their time development. This is consistent with an important role of the internal variability in the precipitation field changes over the past 500 years.

## 1 Introduction

High-resolution regional paleoclimate field reconstructions are able to accurately reproduce the fine spatiotemporal structure of regional climate change on multiple timescales for the period prior to instrumental records. Such reconstructions are an

essential source of information to document the climate variability at decadal to centennial time scales and can be used to assess the ability of climate models to simulate past climate change.

In order to obtain such a reconstruction of regional paleoclimate fields, dense proxy records and an adequate reconstruction method are required. Consequently, climate field reconstructions for the past millennium have focused mainly on regions for which abundant multi-proxy records are available, including Europe (Luterbacher et al., 2004), Northern America (Cook et al., 2004) and East Asia (Shi et al., 2015a). The primary types of proxy records are often tree-ring records and historical documents, mainly because of the data availability and their accurate dating, annual resolution, and demonstrable relationships with instrumental climate data (Fritts, 1976; Zhang, 1991). Other proxy records (e.g., ice core, coral, and varve sediment) have also been introduced into regional climate field reconstructions (e.g., Neukom et al., 2011), but they generally represent a small percentage of the data available in global compilations (e.g. PAGES-2k-Consortium, 2013).

The reconstruction targets are first temperature or temperature-related variables. Reconstructions of the localized precipitation field or other variables related to precipitation are seldom (Cook et al., 1999; Casty et al., 2005; Neukom et al., 2013; Cook et al., 2015a; Seftigen et al., 2015) because they require more dense proxy network that it would be the case for temperature. In particular, the Palmer Drought Severity Index (PDSI) Atlases over the past millennium in North America (North American Drought Atlas, NADA; Cook et al., 1999), in Asia (Monsoon Asia Drought Atlas, MADA; (Cook et al., 2010)), in Europe (Old World Drought Atlas, OWDA; Cook et al., 2015b), and in Oceania (Australia and New Zealand summer drought atlas, ANZDA; Palmer et al., 2015) were reconstructed using the tree-ring records. All datasets are available at the website of National Oceanic and Atmospheric Administration (NOAA) (https://www.ncdc.noaa.gov/data-access/paleoclimatology-data/datasets/climate-reconstruction) and were widely used to identify the variability of droughts and pluvials over the past millennium (e.g., Cook et al., 2015a).

The paleoclimate reconstruction methods are divided into climate index reconstruction (CIR) method and climate field reconstruction (CFR) method according to the reconstruction target. The CIRs are mainly derived from two classes of method, direct regression and indirect regression (Christiansen and Ljungqvist, 2017). Methods using the climate variables as the predictands (or dependent variable) and the proxies as the predictors (or independent variable) are called direct regression. On the contrary, the climate variables as the predictors and the proxies as the predictands leads to indirect regression. The composite plus scale (CPS) method is a widely used, classic direct regression, which composites a group of proxy records using uniform or proxy-dependent weighting. The time series obtained is then scaled to have the same variance as the targeted regional or hemispheric averaged variable over a chosen interval. More generally, the regression process is usually based on some forms of univariate or multivariate linear regression and the regression parameters are estimated using classic methods e.g. ordinary least squares, total least squares, variance matching. However, the problem is generally ill-posed because of the limited number of samples in the calibration period and regularized methods have to be introduced, e.g. truncated principal component regression (truncated-PCR) (Mann et al., 1999), Regularized Expectation Maximization (RegEM) (Mann et al., 2008), and Least Absolute Shrinkage and Selection Operator (LASSO) (McShane and Wyner, 2011).

The local (LOC) method is a promising method based on indirect regression. In the LOC method, each proxy record should be first calibrated using the local instrumental climate data, and the time series are then averaged to obtain the large-scale mean climate index (Christiansen, 2011). An indirect regression is used for the local calibration, justified by the fact that the proxy records are functions of climate variables and not the opposite. The reconstructions based on the LOC method are

assumed to better preserve the low-frequency climate signal compared to other methods, though they would overestimate the high-frequency signal (Christiansen and Ljungqvist, 2011). Then, the optimal information extraction (OIE) method was proposed to address this bias using the arithmetic mean of the regression coefficients of the linear regression and the inverse regression (Shi et al., 2012). The hypothesis in the OIE method is that the regression coefficients are random variables with normal distribution that vary in the ranges between the classic linear regression and inverse regression. Additional methods

have also been proposed recently to take into account some of the biases of classical methods based on regression such as the pairwise comparison (Hanhijärvi et al., 2013) or Bayesian method of various levels of complexity (e.g., Tingley and Huybers, 2010)).

The CFR methods can be divided into the reduced space objective analysis-based method (Evans et al., 2000) and the point-to-point regression-based (PPR-based) method (Cook et al., 1999). A typical example of the first group of method is

provided by the study of Mann et al. (2009) in which the time coefficients of dominant EOF patterns, calculated from instrumental climate data, are estimated over the pre-instrumental data using a network of proxy records, and then, the climate field over the pre-instrumental data is attained from the product of the reconstructed time coefficients and the instrumental dominant EOF patterns. The underlying hypothesis is that the primary spatial modes of climate change during the instrumental period also explain a large fraction of the variability in the past. The advantage of this assumption is that

only a few proxy records with sparse spatial coverage can be enough to reconstruct a climate field (Neukom et al., 2011). However, the discarded EOF patterns after EOF-truncation may retain some small-scale spatial information, which would have been lost. For instance, the global temperature field reconstruction (Mann et al., 2009) was not consistent with a regional temperature field reconstruction in western Qinling Mountains, China (Yang et al., 2013b). In addition, this method is not well adapted for reconstructions of precipitation field because of the high spatial heterogeneity of this variable

(Gómez-Navarro et al., 2015).

The PPR-based method reconstructs each grid point using a linear regression; e.g., PCR (Cook et al., 1999), RegEM regression (Shi et al., 2015a) or the OIE method (Yang et al., 2016) through searching candidate proxy records near the target. The goal of the PPR-based method is to maximizes the retention of spatial information, but this method requires a sufficient number of suitable proxy records near the objective grid points. Selecting only one type of proxy records, with the

associated limited spatial distribution, hinders the field reconstruction of precipitation (or of a variable sensitive to precipitation) for a large-scale region using the PPR-based method. For example, the tree-ring based reconstruction of the MADA provides significant insights into past drought patterns in eastern Asia (Cook et al., 2010), but it performs poorly in reproducing dryness and wetness in eastern China because it only incorporates data from one short tree-ring width chronology from eastern China. Consequently, it would be invalid to extrapolate objective gridded drought variability on the

basis of remote tree-ring records in western China (Yang et al., 2013a). Thus, one method to fully consider regional patterns of precipitation is to use the PPR-based method in conjunction with multi-proxy records with good spatial coverage.

In eastern China, there are abundant historical records of precipitation variability (e.g., Zhang et al., 2003b; Zheng et al., 2006; Hao et al., 2015). Recently, numerous tree-ring records in western China and surrounding regions have been archived in published databases (Yang et al., 2014b; PAGES2k-Consortium, 2017). This presents an excellent and timely opportunity to integrate the data from tree-ring records from western China and historical records in eastern China to reconstruct the precipitation field for the whole of China. Indeed, Feng et al. (2013) reconstructed the precipitation field in East Asia using composites of multi-proxy records, which are widely used to reflect the variability in precipitation (e.g., Liu et al. (2016)). However, the distribution of proxy records in western China was not sufficiently dense in that reconstruction. In addition, spatial information may have been partially lost using the EOF-based method mentioned above. This is a limitation for developing a further understanding of the dominant patterns of precipitation for the period before instrumental records and their possible driving mechanisms. In this paper, we incorporated additional tree-ring records from western China compared to previous studies and used the PPR-based framework with the OIE method to reconstruct the precipitation field for China. We present an empirical attempt to explore the dominant patterns of precipitation variability before the instrumental period and try to analyse their possible origin.

## 2 Data and methods

To reconstruct the precipitation field, we used a gridded instrumental precipitation dataset, proxy records that can be significantly related to precipitation in the domain studied, and the OIE precipitation field reconstruction method. Other datasets were used to validate the reconstruction and explore possible driving mechanisms.

### 2.1 Instrumental data

Owing to the highly heterogeneous and localized variability in precipitation, the accuracy of regional precipitation estimates depends mainly on the spatial density of the stations (Wan et al., 2013). Thus, a dense distribution of datasets is considered a priority.

We selected a monthly gridded precipitation dataset, China's Ground Precipitation 0.5° longitude by 0.5° latitude Grid Dataset V2.0 (Zhao and Zhu, 2015), covering the period AD 1961–2010, since this dataset is calculated based on nearly all available national surface stations (n = 2472) in China. We targeted precipitation data from the warm season (May−September, MJJAS) because historical documents mainly record variations in MJJAS precipitation. The whole of China includes 4189 grid points. The calibration period was set to AD 1961–1990 and the validation period was AD 1991–2000.

Another instrumental precipitation dataset was used in this study to validate the reconstruction: The Homogenized Monthly Precipitation Dataset in China for the interval 1900–2009 (Li et al., 2012), with a 5° longitude by 5° latitude grid resolution. This dataset includes data from all available national surface stations in China before AD 1960. The MJJAS mean

precipitation anomaly for China for the interval AD 1920–1960 is selected here as an independent verification data. Some unusual values were evident for the period before AD 1920, as shown in original Fig. 12 of Li et al. (2012), and these are likely a result of the sparse distribution of observation points. The two datasets are both developed by, and available from, the National Meteorological Information Center, Chinese Meteorological Administration.

## 2.2 Tree-ring record

The use of tree-ring records has multiple advantages, including their annual resolution, easy replication, wide distribution, and significant corroboration from instrumental records. Consequently, such records are considered to be primary and practical archives for reconstructing precipitation fields in China over the past half millennium and are widely used to reconstruct regional precipitation variability in China (e.g., (Shao et al., 2005; Yang et al., 2014b)).

The candidate tree-ring records are required to satisfy three conditions: 1) all records must be archived in public repository; 2) each record needs to at least cover the period from AD 1875 to AD 1977; 3) if raw tree-ring width measurements are available, they must be based on at least five samples for each year to ensure good replication. The screened 372 chronologies include 371 tree-ring width chronologies and a tree-ring oxygen isotope chronology.

To maximize the overlap lengths of the instrument data and proxy records, all tree-ring records were extrapolated to AD 2000 using the RegEM algorithm (Schneider, 2001). Here, the truncation parameters for the RegEM algorithm were set to the same values as that used by Mann et al. (2008). A total of 197 of 372 tree-ring chronologies were extrapolated. The maximum and mean extrapolation lengths of the 197 chronologies were 23 years and 11 years, respectively. The extrapolation bias was ignored because of the short extrapolation length.

We synthesized 372 tree-ring records from China and surrounding area, as shown in Fig. 1. The tree-ring records are located mainly in 14 countries including Bhutan, China, India, Japan, Kazakhstan, Kyrgyzstan, Laos, Mongolia, Nepal, Pakistan, Philippines, Russian Federation, Thailand, and Vietnam. There are only very few tree-ring records in eastern China. This indicates that the currently available tree-ring records are not sufficient for reconstruction of the precipitation field in eastern China.

## 2.3 Dryness/Wetness index (DWI)

In eastern China, abundant records of drought and flood conditions can be found in historical documents, which provide another opportunity to reconstruct past climate (Zhang, 1991; Ge, 2011; Hao et al., 2015). One of the most valuable examples is the Yearly Charts of Dryness/Wetness in China for the Last 500-Year Period dataset (Chinese Academy of Meteorological Science, 1981). The DWI dataset, also known as the drought/flood indices (DFI), has been widely used to assess precipitation variability in eastern China (e.g., Wang and Zhao, 1979; Zhang, 1988; Qian et al., 2003a). This DWI dataset includes data from 120 locations that are distributed mainly in eastern China and northeastern China, with a few in western China. Herein, 13 DWI are excluded because they cover too short time periods. The DWI for each year has five grade values: very wet (grade 1), wet (grade 2), normal (grade 3), dry (grade 4), and very dry (grade 5). The DWI dataset is

mainly derived from the local chronicles and started from 1470 (the sixth Year of Cheng hua reign in Ming Dynasty), which describe the onset, duration, areal extent, and severity of each drought or flood event in each province, China (Chinese Academy of Meteorological Science, 1981; Zhang, 1983). The experts, mainly from the provincial meteorological bureau provinces, China Meteorological Administration, Peking University, and Chinese Academy of Sciences, converted

qualitative textual descriptions into quantitative data (Chinese Academy of Meteorological Science, 1981).

The reliability of DWI was described in previous studies (Zhang, 1983; Zhang, 1988), e.g., the homogeneity of DWI was demonstrated using the chi-square test (Zhang, 1983), and the reliability of DWI in spatial pattern was verified through comparison with the eigenvectors of the instrumental precipitation during the period AD 1951-1974 (Wang and Zhao, 1979). However, DWI have still some weaknesses. The first one is time discontinuity (Zhang, 1983). Figure s1 is the time spans of

107 DWI records. This illustrates that most of DWI records are not continuous in time. The maximum (mean) number of missing values of 107 DWI records during the period AD 1470-2000 was 446 (157). Secondly, the DWI is unevenly distributed over space. Figure s2 shows that the location and number of DWI records during the period AD 1470-2000. On this figure, a value of "100" means that it has 100 observed values during the period AD 1470-2000. It indicates that 91 of 107 DWI records are located in eastern China (east of longitude 105°E), which is the economically developed region. There

are only 16 DWI records west of longitude 100°E. Additionally, an uncertainty due to the subjective judgment is unavoidable for the historical documentary, even though the different sources have been used to cross-validate the final reconstruction (Zhang, 1983; Man, 2009; Ge, 2011; Zheng et al., 2014a). Moreover, the range of values is within five grades and the DFI record is not an accurate precipitation value, thus it also limits the accuracy and ability to detect the extreme events (Zheng et al., 2014a).

In order to improve the quality of the DWI dataset, Professor Zhang De'er lead a team of scientists, that carried out research during 20 years to identify the weather events in China over the past 3000 years. This resulted in the publication of a book, entitled "A compendium of Chinese meteorological records of the last 3,000 years". Each record has been carefully cross-checked from more than 8,000 historical documents (Zhang, 2004). However, the updated dataset is not archived in publish repository so far.

This publicly available DWI dataset has been extended to AD 2000 using the annual average and standard deviation of observed rainy season precipitation (May–September, MJJAS) when the instrumental precipitation is available (Zhang and Liu, 1993; Zhang et al., 2003a). Herein, 85 DWI records need to be interpolated because of missing values. The RegEM method was also used to interpolate the DWI data based on their mutual covariance their mutual covariance with the other available data. The maximum (mean) length for interpolation was 446 (157) years. Any interpolation bias was ignored, as

the historical documentary data record a regional drought or flood event, rather than a local phenomenon, and they show good regional spatial consistency.

In total, we assembled 479 proxy records, which included 371 tree-ring width chronologies, 1 tree-ring oxygen isotope chronologies, and 107 DWI records. Each record is required to be significantly correlated with one or more instrumental precipitation record at the 90% ($p < 0.1$) confidence level during the overlap period, based on both raw data and linearly

detrended data. Moreover, previous studies have repeatedly verified a significant relationship between tree-ring precipitation reconstructions and the DWI on a regional scale (e.g., Zhang, 2010). The common period to all of the proxy records is AD 1875–1977. The number of proxy records has a visible changing point from AD 1470 to AD 1469 after extrapolation/interpolation, with a 41.91% decrease from 136 to 79. The details for each record are provided in Table s1 and Fig. 1.

The two dominant modes of natural climate variability, the El Niño–Southern Oscillation (ENSO) and Pacific Decadal Oscillation (PDO), are used to explore the possible connection between our reconstruction and large-scale variability, since the two indices have already been shown to affect precipitation in China on interannual, interdecadal, and multidecadal timescales based on instrumental analysis (Huang and Wu, 1989; Ma, 2007; Qian and Zhou, 2014). Multiple ENSO and PDO reconstructions over the past millennium have been assembled in our previous work (Shi et al., 2016a). Without loss of generality, we selected three reconstructed ENSO indices (Cook et al., 2008; McGregor et al., 2010; Li et al., 2013a) and three PDO indices (D'Arrigo et al., 2001; D'Arrigo and Wilson, 2006; Shen et al., 2006) that have good performances in relationship with instrumental data. Note that three ENSO indices have a strong significant relationship during the common period AD 1650-1977 with the range of correlation coefficients [0.58, 0.66, 0.84], but the three PDO indices are only very weakly related during their common period AD 1700-1979 with the range of correlation coefficients [0.03, 0.09, 0.13]. As argued recently (e.g., Newman et al., 2016), the PDO cannot be considered as a single dynamical process but results of the combined influence of remote tropical forcing and local North Pacific atmosphere–ocean interactions. This makes it a particularly challenging target for proxy-based reconstructions, explaining the poor agreement between the available series.

## 2.4 Climate model simulation

Six coupled climate models were used to assess whether their past1000 modeling experiments for the interval AD 850–1849 are consistent with our reconstruction. These are BCC-CSM1.1 (Wu et al., 2010b), CCSM4 (Landrum et al., 2012), FGOALS-s2 (Man et al., 2014), GISS-E2-R (Schmidt et al., 2014), IPSL-CM5A-LR (Dufresne et al., 2013), and MPI-ESM-P (Jungclaus et al., 2010). The description of the six models, sponsoring institutions and main references was shown in Table s3 of Shi et al. (2015a). For details and data of the past1000 experiments, see the websites of the Paleo Modelling Intercomparison Project Phase 3 (PMIP3) and the fifth phase of the Coupled Model Intercomparison Project (CMIP5). All simulated results were interpolated to the same temporal and spatial resolution as the reconstruction in this study.

## 2.5 Reconstruction method

The OIE method has been successfully used to reconstruct the South Asian summer monsoon index over the past millennium (Shi et al., 2014), the Northern Hemispheric temperature over the past two millennia (Shi et al., 2015b), and the precipitation field over the past 500 years in western Qinling Mountains, China (Yang et al., 2016). The drawback of this method is an overfitting tendency. An independent test data is needed for cross validation to avoid it.

This method (version 1.3) within the PPR-based framework comprises three steps. The first step is to search for the candidate proxy records. We firstly selected all possible predictors with significant relation to the target. Then, we sorted them as a function of their descending distances from the predictor to the target. We excluded the predictors with distances more than 3500 km. We indeed assumed that the predictor is unlikely to provide useful information about the precipitation variability at the grid point beyond 3500 km. The candidate predictors faced on four possible situations. 1) The number of candidate predictors was more than five, and the first five predictors included at least one record that could reach AD 1470. This target was constructed using the first five predictors. Five is the initial minimum number of candidate predictors according to the past experience (Cook et al., 2013). 2) The number of candidate predictors was more than five, but the first five predictors did not include any record that could reach AD 1470. We increased the number of predictors until a predictor could reach to AD 1470. The target was constructed using the selected predictors, whose number was more than five. 3) The number of candidate predictors is more than five, but no one can reach to AD 1470. We increased the number of predictors until a predictor with the maximum time span. 4) The number of candidate predictors is less than five but more than three. It means that this target cannot meet the requirement of replication, but we tolerate it and use all possible predictors to reconstruct it.

The second step is to determine the weighting for the proxy record using the correlation coefficient between the candidate proxy record and the reconstructed target according to Shi et al. (2014)'s method. The third step is regression of the proxy record using the ensemble LOC regression method (Shi et al., 2012).

Traditional accuracy and skill parameters, including the square of the Pearson product–moment correlation coefficient ($r^2$) between the reconstruction and the instrumental data during the verification period, the reduction of error (RE) in the verification period, and the coefficient of efficiency (CE) in the verification period (Cook et al., 2010), were used to evaluate the reliability of the reconstructions, and the uncertainty was calculated using the standard deviation of the residual between the reconstructed and instrumental precipitation data during the verification period. Moreover, the Pearson's sample linear cross-correlations at lag 0 is used for the correlation analysis, and the significance of the correlation for the filtered time series was accessed using the effective number of degrees of freedom following Zhao et al. (2016).

The ensemble empirical mode decomposition (EEMD) method (Huang and Wu, 2008; Wu and Huang, 2009) was used to analyse the reconstructed mean MJJAS precipitation time series for eastern China, western China, and the whole of China. The eastern and western China is simply divided along - the longitude 105°. Following Mann et al. (1995), the interannual timescale was set to < 8 years. The interdecadal timescale was defined as ≥8 years and <35 years. The multidecadal timescale was defined as ≥35 years and <100 years, and the centennial scale was >100 years.

The superposed epoch analysis (SEA) is traditionally used to analyse the influence of volcanic eruption on the climate, e.g., Bradley (1988). Here, the code to compute SEA has been downloaded from the website (http://blarquez.com/superposed-epoch-analysis-sea/). The period analysed (time window) are set as 20 years before and after each volcanic eruption event. The 90% confidence limit is estimated using the bootstrap procedure (Blarquez and Carcaillet, 2010). The eruption time series of Sigl et al. (2015) is used here because of the dating improvement compared to earlier estimates. Four categories of

volcanic eruption events during the period from AD 1490 to AD 1829 are chosen following Zhuo et al. (2014)'s method which is based on the magnitude of their sulfate deposition in the Greenland ice-core records: (1) all Northern hemisphere eruption events (CNH0P) according to Sigl et al. (2015), (2) CNH1/2P: the eruption events that have more than half, (3) equal (CNH1P), and (4) double (CNH2P) that of the sulfate deposition of the 1991 Mount Pinatubo eruption.

**3 Results and discussion**

The quality and reliability of the reconstruction are illustrated in Figures 2−4. Firstly, the spatial distribution of the number of predictors for each grid is shown in Fig. 2a. As mentioned above, the initial minimum number of predictors is five. The 2599 grids with five candidate proxy records account for 62.04% of the grids. The 3760 grids with ≤10 candidate proxy records account for 89.76% of the grids. The maximum number of predictors is 38. Only seven grids cannot be satisfied with

the initial minimum number of predictors (five). This indicates that 99.86% grids have good replication of the reconstruction. In order to reconstruct these seven grids, we tolerate the four grids with four predictors, and the three grids with three predictors. This means that these seven grids were not followed the requirement of the replication. Secondly, the maximum distance from the predictor to the target for each grid is shown in Fig. 2b. The 1599 grids with a ≤450km search radius account for 38.17% of the grids. The 3467 grids with search radii of ≤1500 km account for 82.76% of the grids. The

maximum distance to the target is 3495.5 km. This implies that precipitation in most of the grid points can be reconstructed using nearby proxy records. Thirdly, Figure 3 presents a summary of the reconstruction skills. Figures 3a−c show that the similarity in the patterns among the r2, the RE and the CE maps, characterized by a better quality of the reconstruction in eastern China (with the exception of some regions in northeastern China) than in western China. The maximum explained variance is 0.96. The number of grids for which both the RE and CE values are greater than zero is 3631, accounting for

86.68% of the grids. This indicates that most of the grid points pass the cross-validation process. The uncertainties associated with the grids in southeastern China are greater than those for the grids in northwestern China in Fig. 3d because of large precipitation anomalies in southeastern China.

Finally, Figure 4a compares the reconstructed MJJAS mean precipitation anomalies with the instrumental MJJAS mean precipitation anomalies (Zhao and Zhu, 2015) for China for the interval AD 1961-2000. The reconstructed MJJAS mean

precipitation anomalies mostly agree with the instrumental data. The correlation coefficient is 0.89 (n = 40), which is significant at the 99% confidence level. Figure 4b compares the reconstructed MJJAS mean precipitation anomalies with the instrumental MJJAS mean precipitation anomalies (Li et al., 2012) in China during AD 1900-2000. The reconstructed MJJAS mean precipitation anomaly is significantly correlated to the instrumental independent data during the interval AD 1920-1960, with a correlation coefficient of 0.59 (n = 41), also significant at the 99% confidence level. This indicates that

the reconstruction passes the out-of-sample validation on the mean MJJAS precipitation anomalies for China. A part of the disagreements before AD 1919 can come actually from the uncertainties of Li et al. (2012), as explained in the instrumental data section.

The evolution of regional mean precipitation anomalies is exhibited in Fig. 5. The different components of the MJJAS precipitation anomalies for eastern China (east of 105°E), western China (west of 105°E), and whole China over the past 531 years (AD 1470-2000) are obtained using the EEMD method (Fig. 5 and Fig. s4). The amplitudes of interannual and interdecadal components in eastern China are much larger than in western China (Fig. s4), but the differences of the amplitudes of other components between eastern and western China are less clear in Fig. 5. The drought/flood changes in eastern and western China are generally consistent over the multidecadal time scale in Fig.5a.

The centennial components in eastern and western China describe both a relative wet climate during the 16th century and a drought during the 17th century. The 17th century drought is also reported in previous studies (e.g., Wang et al., 2002). The correlation coefficient of the centennial component in eastern and western China during the interval AD 1470-1749 is 0.87, but the correlation coefficient during the interval AD 1750-2000 is -0.62. Moreover, the 101-year running correlation between the centennial components in eastern and western China shows that a strong significant positive relationship gradually changed to a weak significant negative relationship during the late 18th century (figure not shown). This may suggest that the driver of the centennial component has changed after the late of 18th century.

The long-term trends in eastern and western China have opposite signs. The long-term trend in eastern China can be broadly divided into two periods: during the first phase before the early 18th century there was a wetting trend, and then, the conditions become dryer until now, which is consistent in previous studies (e.g., Zheng et al., 2006; Pei et al., 2015). The long-term trend in western China can also be divided two stages: during the first stage from the late 15th century to the late 16th century there was a drying trend, and then, the second stage corresponds to gradually wetter conditions until now. The long-term trend for whole China has similarities with the one in eastern China, but with a much weaker amplitude.

Figure 6 shows the spatial patterns of the May–September precipitation field relative to the 1961–90 climatological mean during five severe droughts in China. The selection of five drought periods follows Feng et al. (2013) but the spatial patterns in our reconstruction display some clear differences from their results. A "north drought with south flooding" dipole pattern in eastern China is seen in Figures 6a−d, and there is a triple pattern in eastern China in Fig. 6e. A similar dipole pattern can be found in Figures 5b and 5d (Feng et al., 2013) for two drought events (AD 1586−89 and AD 1876−78) but during four out of five drought periods, most of northeastern China is relatively humid situation in our study, which is not consistent with Feng et al. (2013)'s Fig. 5. An exceptional drought condition in northeastern China appears in the 1876-1878 drought event, which is the most severe drought of five events in eastern China.

We have also calculated the spatial correlation between Cook's MADA PDSI reconstruction (Cook et al., 2010) with our precipitation reconstruction in Fig. s3. Strong correlations appear in the northeastern Tibetan Plateau, where there are longest and most abundant tree-ring width chronologies in China. Here, the precipitation is possibly a primary control factor of the tree-ring width chronology (Zhang et al., 2003c; Yang et al., 2014b). There are weak correlations between the reconstructions in eastern China even through the correlation coefficients are significant in some regions. The MADA is not consistent with the DWI records in eastern China, since only very few and short tree-ring width chronologies in eastern

China are used to reconstruct the MADA (Yang et al., 2013a; Kang et al., 2014; Yang et al., 2014a; Zheng et al., 2014b; Ge et al., 2016).

We compared the precipitation reconstruction with the six climate model simulations in Figures 7−8. We start our analysis in AD 1470 since, before that date, there are no available historical document and the quality of the reconstruction is thus likely low. The analysis of spatial pattern covers the period AD 1470-1849 as we specifically focus on the pre-industrial period. The more recent past has been the subject of some recent studies and different factors such as the changes in aerosol concentration may have a dominant effect then (Li et al., 2016). Figure 7 shows the reconstructed 9-year running mean MJJAS precipitation anomalies for China for the interval AD 1470–2000 compared with six climate model simulations. Only the CCSM4 results are significantly correlated with the reconstructed results during AD 1470–1849 at the 95% confidence level, but the correlation is low negative (r = -0.16). Furthermore, there is distinct shift between the mean values of the CCSM4 results and the reconstruction over that period. This is related to the choice of the reference period and the increasing trend in FGOALS-s2 model since the mid-19$^{th}$ century. All climate model simulations show low correlations with the reconstructed result. The correlation of the simulated times series is also weak between the different models. This suggests that MJJAS mean precipitation anomalies over the past 380 years in China may be largely controlled by the internal variability rather than by external forcing during the interval (AD 1470–1849). Similar conclusions were derived from the comparison of reconstructed and simulated hydroclimatic variables over the past millennium in North America (Coats et al., 2015) and in East Africa (Klein et al., 2016).

Traditional EOF analysis was applied to reveal the spatial patterns in MJJAS precipitation anomalies in China over a 380-year interval (AD 1470–1849). The first four EOFs are well separated according to North et al. (1982) criteria, but the fourth EOFs only accounts for 4.7% of total variance, and its pattern is unusual compared to previous studies. Thus, the first three EOF patterns and their corresponding time coefficients (also known as principal components, PCs) of the reconstructed MJJAS precipitation fields in China are compared with six climate model simulations in Fig. 8.

The first EOF leading mode of the reconstructed MJJAS precipitation field (Fig. 8a) displays a main loading in eastern China, and a general (monopole) variation over most of China, with the exception of the northeastern and western margins of the Tibetan Plateau. This mode accounts for 16.6% of the total variance, which is lower than the leading mode of temperature field (Shi et al., 2015a), but it is normal in precipitation analysis (Day et al., 2015). A main loading in eastern China also appears in the EOF1 of the reconstructed MJJAS precipitation anomalies during the interval (AD 1850–2000), the EOF1 of the reconstructed data during the interval (AD 1961–2000) and the EOF1 of the instrumental data during the same interval (AD 1961–2000). A main loading of EOF1 in eastern China is also consistent with other previous results (Wang and Zhao, 1979; Qian et al., 2003a). This is due to the large variance in eastern China, which causes the larger loads in eastern China in the other two EOFs. The EOF1 of IPSL-CM5A-LR, the EOF2 of MPI-ESM-P, and the EOF3 of CCSM4 also show a consistent variation in most of eastern China, but also some differences with the pattern deduced from the reconstruction. Furthermore. the corresponding time coefficients of model EOFs shows no obvious significant relationship with the reconstructed data (figure not shown), which was expected if natural variability is the main driver of the changes.

The second leading mode of the MJJAS precipitation field (Fig. 8b) demonstrates a south–north anomalous rainfall dipole pattern, with drying in the middle and lower reaches of the Yellow River, and increasing rainfall across and to the south of the Yangtze River. This mode accounts for 11.2% of the total variance and also appears in the EOF2 of the reconstructed MJJAS precipitation anomalies during the interval (AD 1850–2000). The South-Flood North-Drought pattern is commonly referred in previous studies from an analysis of DWI (e.g., (Wang and Zhao, 1979; Qian et al., 2003a) and instrumental data (e.g., , (Huang et al., 1999;Yu and Zhou, 2007;Ding et al., 2008;Zhou et al., 2009). Moreover, the variations have the same sign in most of western China and northern China, except for Xinjiang province. The EOF1 from three climate models (CCSM4, FGOALS-s2, GISS-E2-R), the EOF2 from BCC-CSM1.1 model, and the EOF3 from MPI-ESM-P model reproduce a similar south–north dipole pattern in eastern China to the reconstructed results, but the specific range for each model is different. Moreover, their corresponding time coefficients show that no climate model simulation demonstrates a significant relationship with the reconstructed result (figure not shown). As mentioned for EOF1, this may be perfectly well justified if the variability of the south–north dipole pattern is dominated by internal variability.

The third leading mode illustrates a "sandwich" triple precipitation pattern with increasing rainfall in the area covering the middle and lower reaches of the Yangtze River valley, drying over southern and northern China, and variations of the same sign in most of western China and central China (Fig. 8c). This mode accounts for 7.9% of the total variance. The "sandwich" triple mode in eastern China has been reported based on the analysis of DWI (e.g., (Wang and Zhao, 1979; Qian et al., 2003b)) and instrumental data (e.g., (Ding et al., 2008)). The EOF1 from two climate models (BCC-CSM1.1 and MPI-ESM-P), the EOF2 from CCSM4 model, the EOF3 from IPSL-CM5A-LR model show similar "sandwich" triple mode to the reconstruction, and their corresponding time coefficients have again no significant relationship with the reconstructed result at the 90% confidence level (figure not shown).

In order to explore the origins of three dominant modes, we firstly consider the influence of the external forcing on the MJJAS precipitation anomalies variability during AD 1470-1849. The impact of the Northern Hemisphere volcanic eruptions on the precipitation field for the four categories of eruption (CNH0P, CNH1/2P, CNH1P, and CNH2P) is shown in Fig. s5. The SEA results applied to the mean precipitation anomalies (Fig. s5a), and its PC1 (Fig. s5b) during AD 1490-1829 shows that volcanic activity as one important external forcing might affect the MJJAS precipitation anomalies variability for China. Nevertheless, the signals are barely significant and there are similar averaged scores before and after the volcanic eruption year. Moreover, the spatial pattern of the impact of the Northern Hemisphere volcanic eruption events on the precipitation field (Figures s5c−f) is not consistent between the four categories of eruption (CNH0P, CNH1/2P, CNH1P, and CNH2P). This indicates that the response of MJJAS precipitation anomalies for China to Northern Hemispheric volcanic eruption is not robust.

The solar activity, as another potentially important external forcing, may also be part of the driving mechanism. This view is supported by the fact that the PC1 shows a weak significant relationship with solar activity index (Wang et al., 2005) ($r = 0.19$, $n = 240$) at the 95% confidence level for the interval AD 1610–1849. The correlation coefficient reaches 0.35 after the 11-year running mean filter. In summary, the influences of volcanic eruption and solar activity on PC1 are not very strong in

our results. A pattern showing some similarities to the PC1 of the reconstructions appears in three climate models (Fig. 8a), but the differences, in particular in western China, are too large to ensure that it has the same dynamical origin and to use model results to determine the origin of the reconstructed pattern.

The second mode of annual precipitation field is the north–south dipole mode in eastern China, with variations of the same sign in most of western China and northern China, except for Xinjiang province. In fact, the north-south dipole pattern of the precipitation in eastern China was found over centennial timescale during the Medieval Warm Period and the Little Ice Age from the historical documents and speleothem records (e.g., (Wang et al., 2001) and was one of dominant modes over interdecadal timescale (e.g., Ding et al., 2008).

In order to explore its possible origin, we have calculated the running correlation between the 17 reconstructed ENSO indices and the precipitation anomalies averaged over Yangtze River region and Huai River region with a window size of 101 years (Figures not shown). The correlation varied between negative and positive values, but all show small and positive correlation, when we have calculated the correlation coefficients for the full period. This indicates that the relationship between ENSO and summer precipitation in China is not stable in time, which is consistent with the instrumental period (Wu and Wang, 2002). However, if we analyse the full period, a more robust link between ENSO and summer precipitation in China may be found.

We calculated the correlation of the precipitation field with the annual mean (over the months July–June) ENSO index of McGregor et al. (2010), as shown in Fig. 9. The results show a similar pattern with a north–south dipole mode in eastern China, and the precipitation anomalies in most of western China have a positive correlation with ENSO at the 90% confidence level. In addition, PC2 is significantly correlated with the ENSO index reconstruction (McGregor et al., 2010) at the 99% level (r = 0.30, n =200) during the interval AD 1650–1849. Moreover, two other ENSO indices (Cook et al., 2008; Li et al., 2013a) give similar correlation maps with the precipitation field (Fig. s6), but a lower correlation coefficients with PC2. This indicates that the north–south dipole in eastern China and variations of the same sign in most of western China and northern China, except for Xinjiang province, are likely influenced by ENSO variability before the Industrial Revolution in our reconstruction.

Finally, we calculated simulated Niño 3.4 in different seasons including the annual mean (over the months July–June), previous July to current June, previous December–January–February (DJF), current March–April–May (MAM), current June–July–August (JJA) and current MJJAS seasons. The correlation maps of five simulated MJJAS mean precipitation anomalies for China with the five-corresponding simulated annual mean Niño 3.4 indices are shown in Fig. 9. They display similar south-north dipole correlation patterns in eastern China, similar to the one from the reconstruction, for three climate models (BCC-CSM1-1, CCSM4, and MPI-ESM-P). The relationship between the previous winter (December–January–February) Niño 3.4 index and the precipitation field for each model is shown in Fig. s7. The spatial pattern is very similar with the result of annual mean Niño 3.4 index. The Niño3.4 indices in previous July to current June, previous December–January–February (DJF) and current March–April–May (MAM) seasons during AD 1470-1849 in CCSM4 model are significantly related to its PC1 at the 99% confidence level, and the correlation coefficients are 0.30, 0.30 and 0.27,

respectively. The Niño3.4 indices in current June–July–August (JJA) and MJJAS seasons in FGOALS-s2 model during AD 1470-1849 are significantly related to its PC1 at the 99% confidence level, and the correlation coefficients both are 0.16. The Niño3.4 indices in previous July to current June, previous DJF, current MAM, JJA and MJJAS seasons during AD 1470-1849 in MPI-ESM-P model are significantly related to its PC3 at the 99% confidence level, the correlation coefficients are 0.23, 0.22, 0.20, 0.18, and 0.19, respectively. The EOF1 of CCSM4 model, the EOF1 of FGOALS-s2 model and the EOF3 of MPI-ESM-P model all show a similar south-north dipole mode, even the specific ranges of their spatial patterns are different. This demonstrates that ENSO has likely an imprint on the south-north dipole mode of the precipitation pattern in eastern China during AD 1470-1849 in those simulations.

Those results are consistent with previous studies based on PMIP3 model simulations suggested that La Niña (El Niño)-like conditions may explain the north-south dipole in eastern China on centennial timescale (e.g., Shi et al., 2016b) and with instrumental observations indicating that ENSO was associated with summer rainfall in eastern China (e.g., Huang and Wu, 1989; Guo et al., 2012; Schubert et al., 2016). Based on instrumental data analysis, three general views, are used to explain the precipitation variability in eastern Asia linked to ENSO, and all of them emphasize the important bridge role of the anomalous western North Pacific anticyclone. The first one is the equatorial Rossby wave response to ENSO via the Pacific-East Asia teleconnection (Wang et al., 2000; Zhang et al., 2011; Karori et al., 2013; Feng et al., 2016). The second one is equatorial Kelvin wave response to Indian Ocean warming during El Niño decaying summer which is named "Indian Ocean capacitor effect" (Xie et al., 2009). The more recent third one is the nonlinear atmospheric interactions between ENSO and the annual cycle (Stuecker et al., 2013; Zhang et al., 2016).

We calculated the correlation map of the precipitation field with the PDO index (D'Arrigo et al., 2001) applying a 9–year running mean filter (Fig. s2). The results at the 90% confidence level show a pattern similar to EOF2 with a north–south dipole mode in eastern China. Moreover, the relationship between PDO index (D'Arrigo et al., 2001) and PC2 is strongly significant (r = 0.41, n = 150) during AD 1700–1849 at 95% confidence level after a 9-year running mean filter. However, the other two PDO indices (D'Arrigo and Wilson, 2006; Shen et al., 2006) give different correlation maps with the precipitation field (Fig. s2), and lower correlation coefficients (-0.25 and 0.36) with PC2 after a 9-year running mean filter. This indicates that the EOF2 mode is also possibly related to variations in the PDO, but the result is sensitive to the choice of the reconstructed PDO index.

Based on the instrumental data analysis, the "sandwich" triple mode in eastern China is likely associated with a meridional tripolar teleconnection in eastern Asia: the Pacific-Japan (PJ; (Nitta, 1987)), Pacific-East Asia (EAP, (Huang and Li, 1988)), or Indo-Asia-Pacific (IAP; (Li et al., 2013b)). The PJ/EAP/IAP teleconnection pattern can be considered as an internal mode mainly controlled by atmospheric processes (Hirota and Takahashi, 2012; Zhang and Zhou, 2015). It also can be forced by the external heating such as the anomalous convective activity in the western Pacific and tropical Indian Ocean during the El Niño decaying year (Huang and Li, 1988; Li et al., 2013b; Xie et al., 2009; Wu et al., 2010a). However, there is no district evidence from the correlation maps of the reconstructed precipitation field with the ENSO and PDO indices to support this kind of mechanism for the "sandwich" pattern in this study. Alternatively, a new hypothesis was proposed recently to

explain the "sandwich" triple mode through the interannual change in the strength of moisture transport from the Bay of Bengal to the Yangtze corridor across the northern Yunnan Plateau (Day et al., 2015). The increased latent heating associated with an increase in water vapor along the Yangtze corridor may generate the triple mode in eastern China and variations of the same sign in most of western China and central China.

Our results indicate thus that the south-north mode variability of precipitation anomalies in China carries very likely the fingerprint of ENSO evolution in tropical Pacific over the past 500 years. The origin of the EOF1 and EOF3 patterns over the pre-industrial period is not clearly established yet, even though both of them maybe related to the movement and intensity of the western pacific subtropical high during the instrumental period (Wu and Wang, 2002). Moreover, some studies show that other factors such as the North Atlantic Oscillation (NAO) (Wu et al., 2009; Zheng et al., 2016) and the
North Atlantic triple SST pattern (Ruan and Li, 2016), the interdecadal Pacific oscillation (IPO) (Song and Zhou, 2015), the snow cover change of the Tibetan Plateau (Ding et al., 2009; Wu et al., 2012), and some regional processes in China may contribute to the precipitation field modes during the instrumental period. Thus, additional studies are then required to determine which of these processes might be related to EOF1 and EOF3 over the pre-industrial period.

Some climate models (e.g., CCSM4, MPI-ESM-P) can broadly reproduce some of the dominant spatial patterns of variability
of the reconstructed precipitation field for period studied. Nevertheless, the corresponding time coefficients do not match with the reconstructed series. A possible reason for this is that the precipitation changes are controlled by internal variability (e.g., related to ENSO). By constraining model result to follow the observed time series, data assimilation may then provide an interesting opportunity to analyse in more detail the mechanisms at the origin of the reconstructed changes (e.g., (Widmann et al., 2010; Hakim et al., 2016)).

**4 Conclusions**

The precipitation field for all of China was reconstructed for the past half millennium using the OIE method and additional proxy records compared to previous studies. The reconstruction shows good performance through the cross-validation process and comparison with "out-of-sample" instrumental data.

The precipitation field reconstruction reveals three leading modes for the period AD 1470–1849 before the Industrial
Revolution. The first dominant mode shows consistent variation across most of China, with the exception of the northwestern Tibetan Plateau and some area of the Xinjiang province. This mode does not appear to be associated to the response to volcanic eruption or the solar activity. A hypothesis is that such homogenous precipitation variations in various climate regions in China have their origin in the internal variability of the system but it was not possible to determine in the present framework through which mechanism. The second mode, comprising a north–south dipole in eastern China and
variations of the same sign in most of western China and northern China, except for Xinjiang province. The correlation with different reconstructions of ENSO index indicates that this dipole is likely related to variations in ENSO. The third mode is a "sandwich" triple mode in eastern China and variations of the same sign in most of western China and central China.

Moreover, the precipitation field reconstruction was used to assess the skill of PMIP3 coupled climate models. For most models, the dominant mode of variability is not characterized by relatively homogenous changes over all China, in contrast to the reconstructed fields. The correlation map between the five simulated MJJAS mean precipitation anomalies for China with the five-corresponding simulated annual mean Niño 3.4 indices shows that the ENSO has likely an imprint on the south-north dipole mode of precipitation anomaly in eastern China over the half past millennium in the simulations too. However, there is a clear model-reconstruction mismatch in reproducing the corresponding time development as they are not able to reproduce the timing of events associated to internal variability.

**Data availability**

The 479 proxy records include 247 tree-ring width chronologies and a tree-ring δ18O chronology archived in International Tree Ring Data Bank (ITRDB) (https://www.ncdc.noaa.gov/data-access/paleoclimatology-data/datasets/tree-ring), 77 tree-ring width chronologies in the supplement of the book (Li et al., 2000), 37 tree-ring chronologies in the PAGES2k_dataset v2.0 (https://figshare.com/s/d327a0367bb908a4c4f2), 10 tree-ring width chronologies and 107 Dryness/Wetness records at the webpages of the Chinese Meteorological Data Service Center (CMDC) (http://data.cma.cn/data/detail/dataCode/HPXY_HDOC_CHN_DAW.html, and http://data.cma.cn/data/cdcdetail/dataCode/HPXY_TRRI_CHN.html). The two instrumental precipitation datasets are both archived by the Chinese Meteorological Data Service Center (CMDC). The China's Ground Precipitation Dataset V2.0 can be obtained from the website (http:// data.cma.cn/data/detail/dataCode/SURF_CLI_CHN_PRE_MON_GRID_0.5.html) and the Homogenized Monthly Precipitation Dataset in China can be downloaded from the website (http://data.cma.cn/data/detail/dataCode/SEVP_CLI_CHN_PRE_ MON_GRID.html). We will archive the code for the OIE (version 1.3) method on GitHub website after this work published.

**Acknowledgements**

This work was jointly funded by the National Key R&D Program of China, Ministry of Science and Technology of the People's Republic of China (Grant No. 2016YFA0600504), and the National Natural Science Foundation of China (Grants No. 41505081, No. 41690114, and No. 41430531). Feng Shi was supported by the "MOVE-IN Louvain" Incoming Post-doctoral Fellowship, co-funded by the Marie Curie actions of the European Commission. Hugues Goosse is Research Director with the FNRS-Fonds de la Recherche Scientifique, Belgium. We thank François Klein for his help in the processing of the data. Thanks are extended to Kevin Anchukaitis, H. P. Borgaonkar, Achim Bräuning, Brendan Buckley, Edward R. Cook, Zexin Fan, Keyan Fang, Narayan P. Gaire, Xiaohua Gou, Minhui He, Katsuhiko Kimura, Paul J. Krusic, Jiangfeng Li, Jinbao Li, Hongbing Liu, Chun Qin, Jonathan Palmer, Tatyana Papina, Jianfeng Peng, Somaru Ram, Masaki Sano, Margit Schwikowski, Xuemei Shao, Paul Sheppard, Jiangfeng Shi, Shri A.B. Sikder, Olga Solomina, Jianglin Wang,

Bao Yang, Koh Yasue, Yujiang Yuan, Muhammad Usama Zafar, Masumi Zaiki, and other paleoclimatic scientists who published the various tree-ring chronologies used in this study.

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

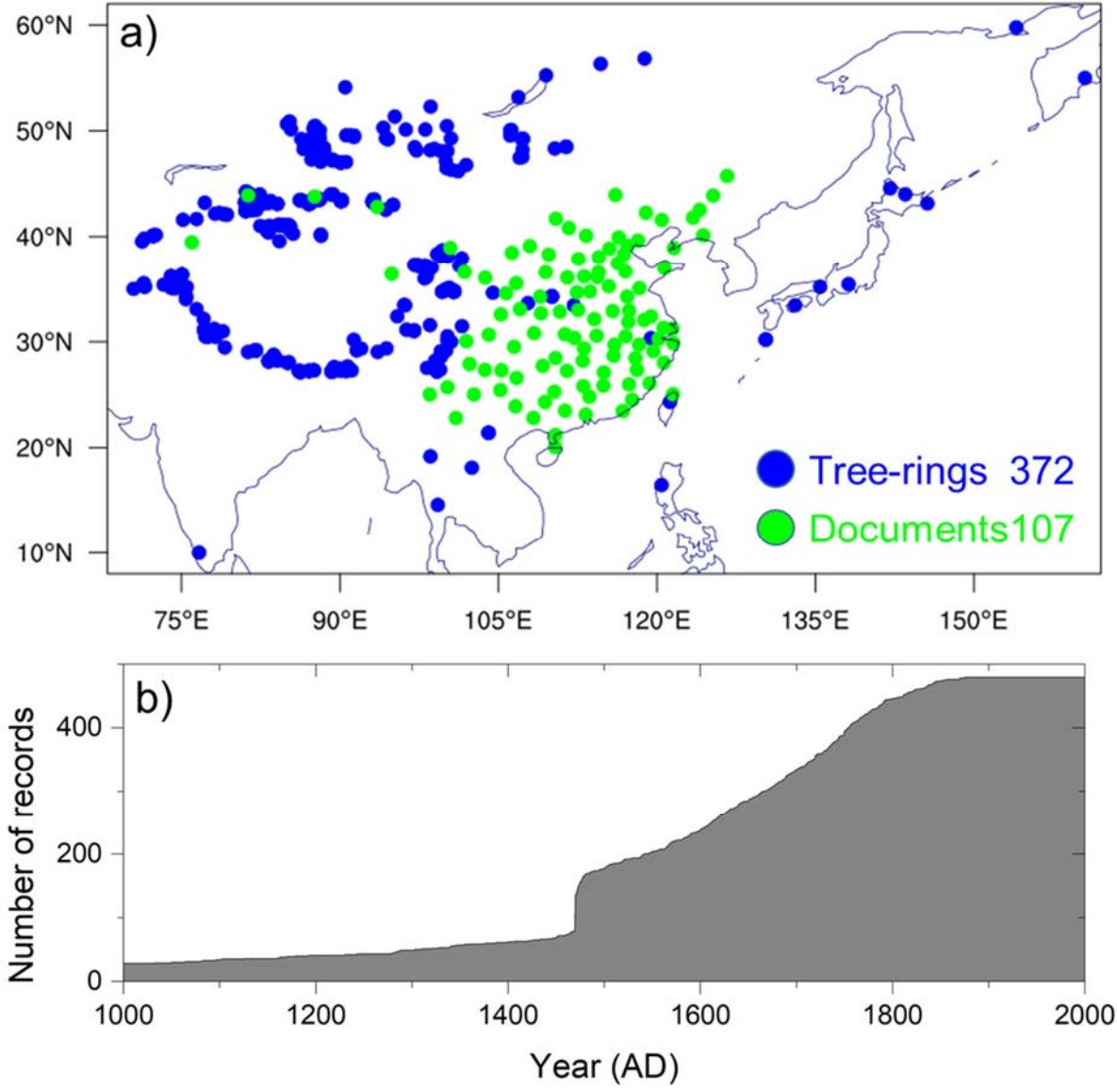

**Figure 1. Map showing the locations of proxy records (a) and plot of the number of proxy record for each year (b).**

**Figure 2.** The spatial distributions of the number of proxy predictors (a) and the maximum distance from the predictor to the target (b) for each grid point.

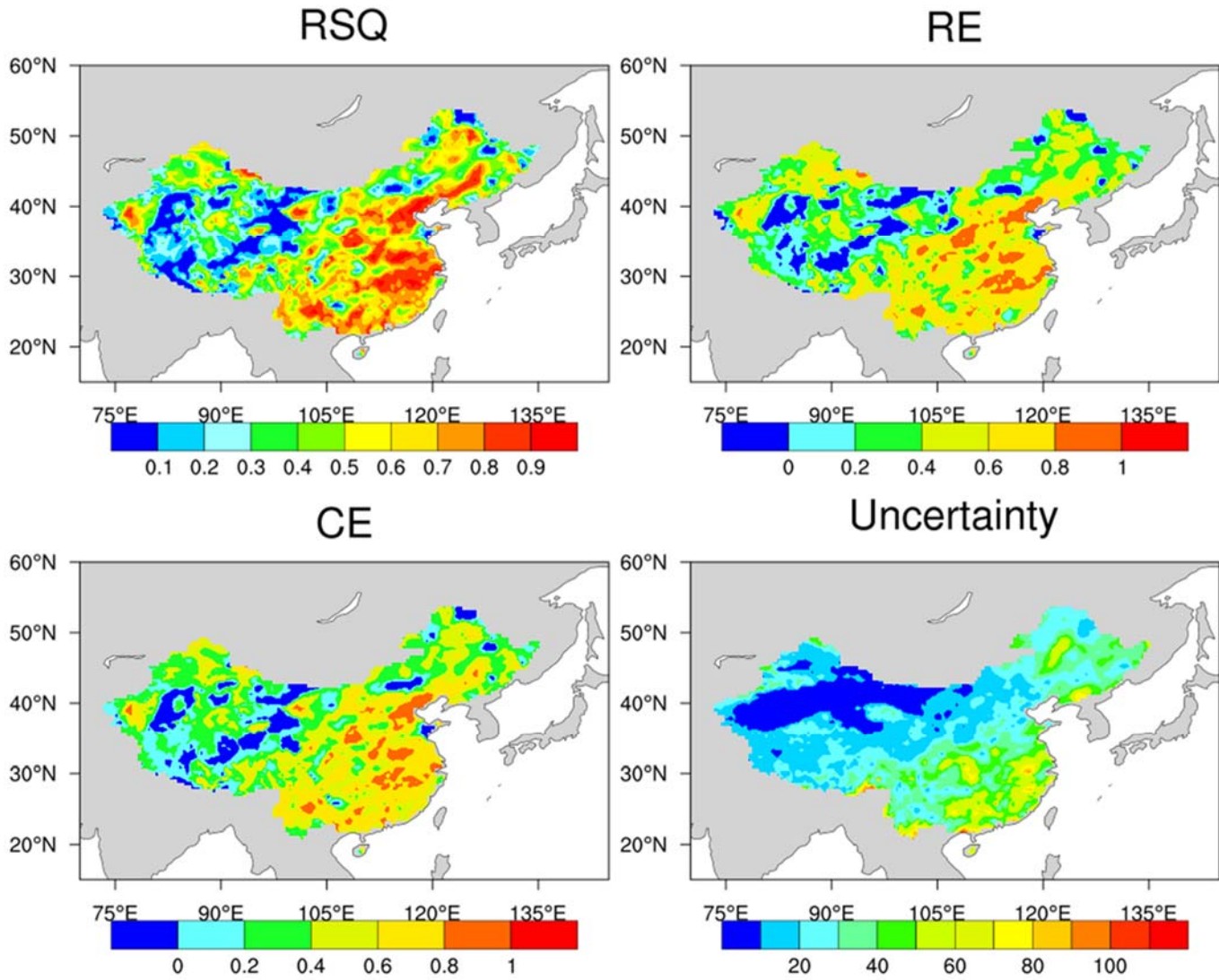

**Figure 3. Skills of the reconstructed MJJAS mean precipitation anomalies in China for the 1961–1990/1991–2000 calibration/verification period. The $r^2$ is the square of the Pearson product–moment correlation coefficient, the RE and CE are the reduction of error and the coefficient of efficiency, the uncertainty is characterized from the standard deviation of the residual between the reconstructed and instrumental precipitation data during the verification period.**

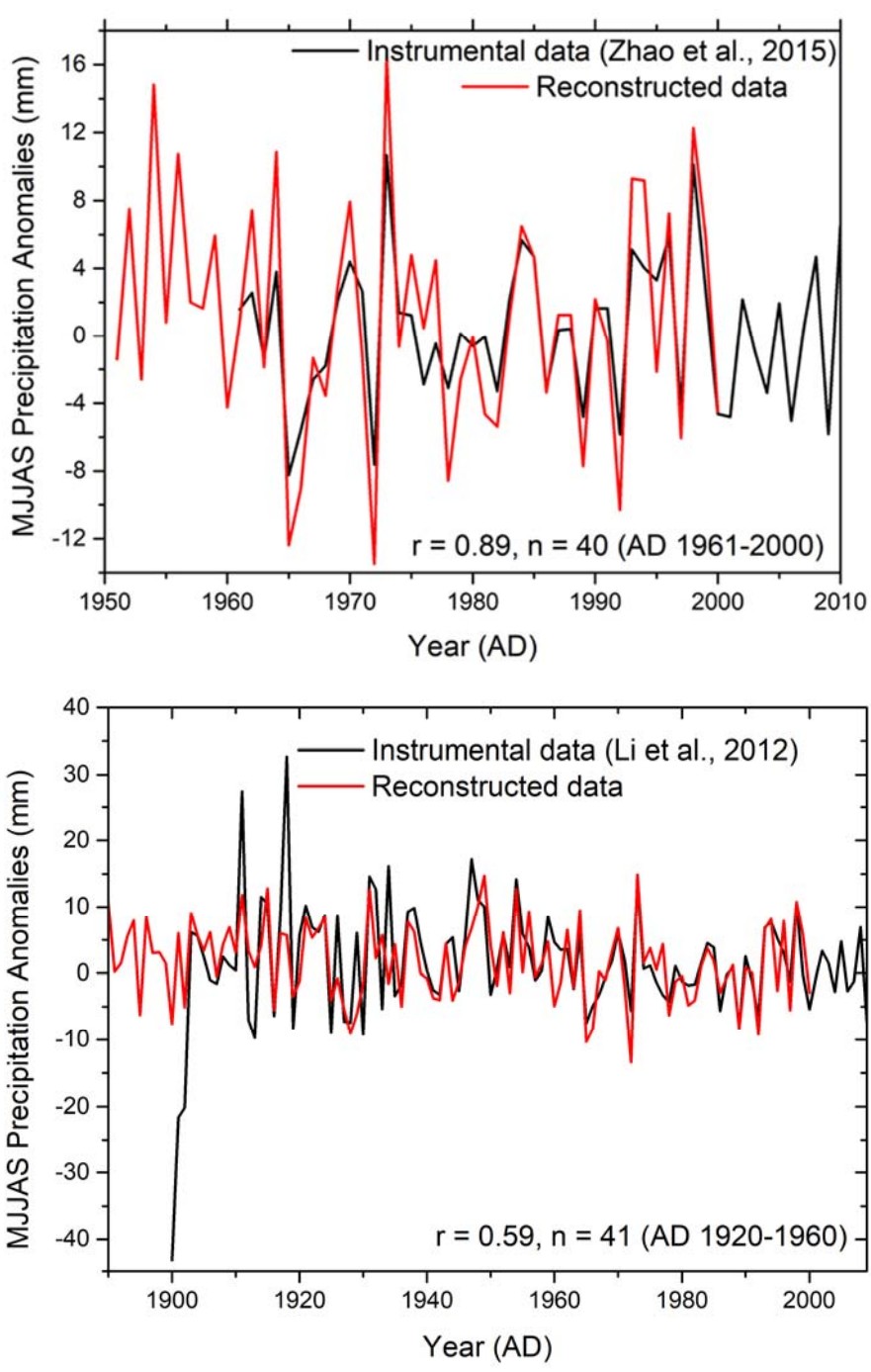

**Figure 4. Comparison of the reconstructed and instrumental MJJAS mean precipitation anomalies for China. (a) instrumental data (Zhao et al., 2015); (b) instrumental data (Li et al., 2012).**

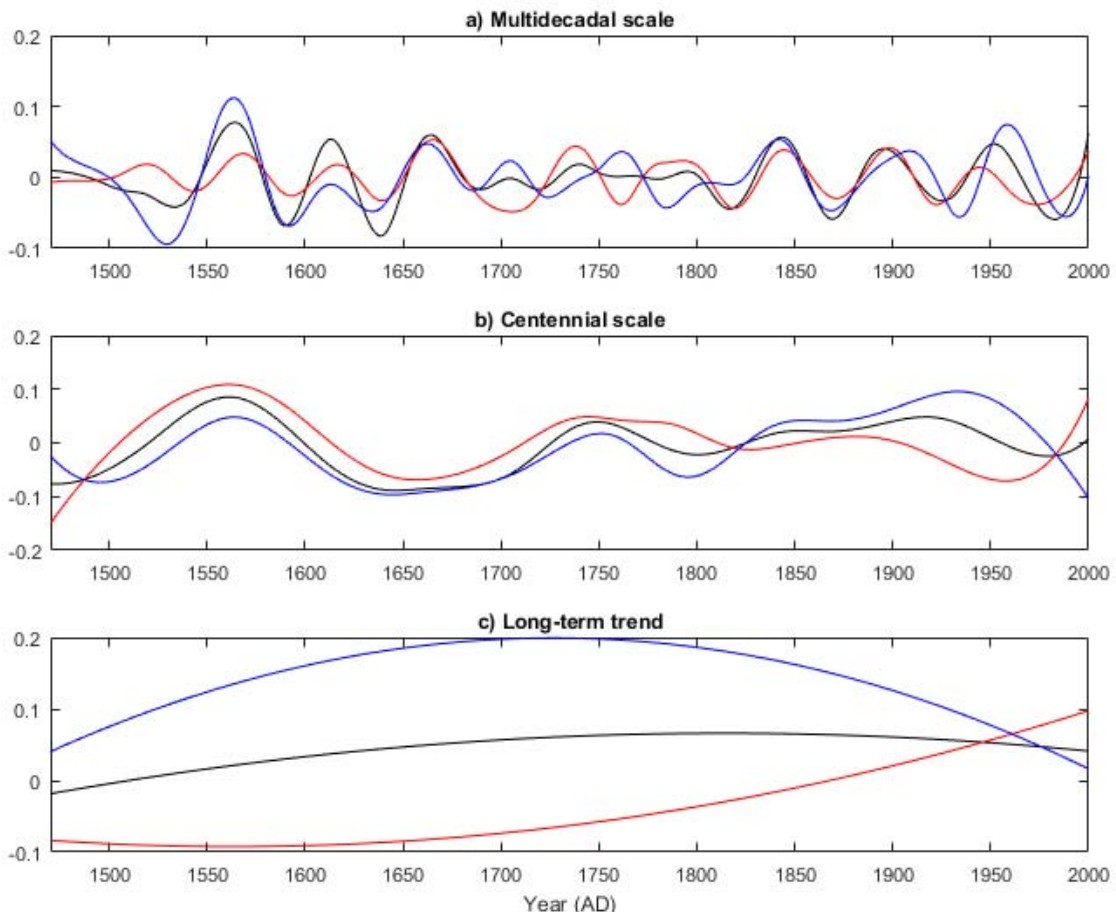

**Figure 5. The intrinsic mode functions (IMFs) of the mean MJJAS precipitation anomalies for eastern China (blue line), western China (red line), and whole China (black line), over the past 531 years (AD 1470-2000) using the Ensemble Empirical mode decomposition (EEMD) method, including the multidecadal component (a), centennial component (b), and long-term trend (c).**

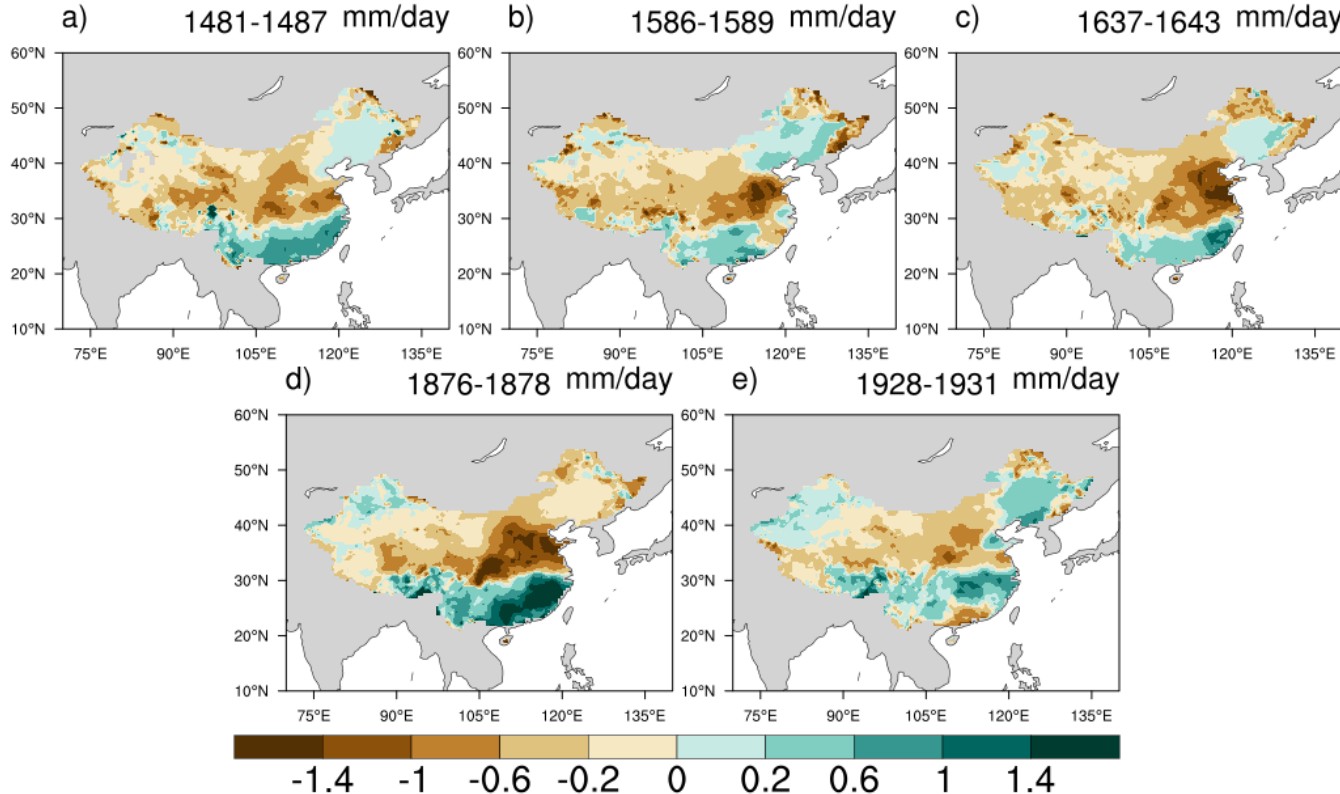

**Figure 6: Spatial patterns of the May–September precipitation field relative to the 1961–90 climatological mean during five severe droughts in China.**

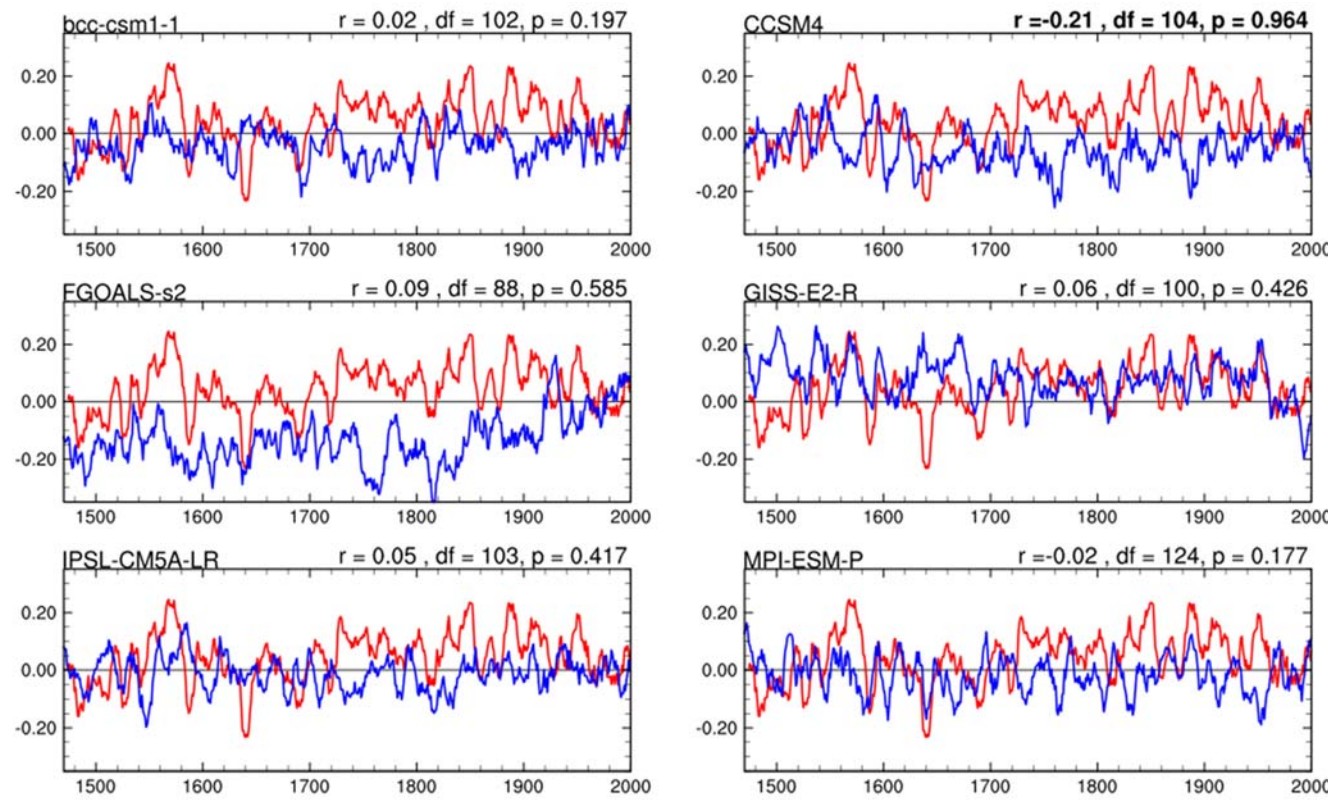

**Figure 7. Comparison of the 9-year running mean reconstructed (red line) and six simulated (blue line) MJJAS mean precipitation anomalies for China during the interval AD 1470–2000.**

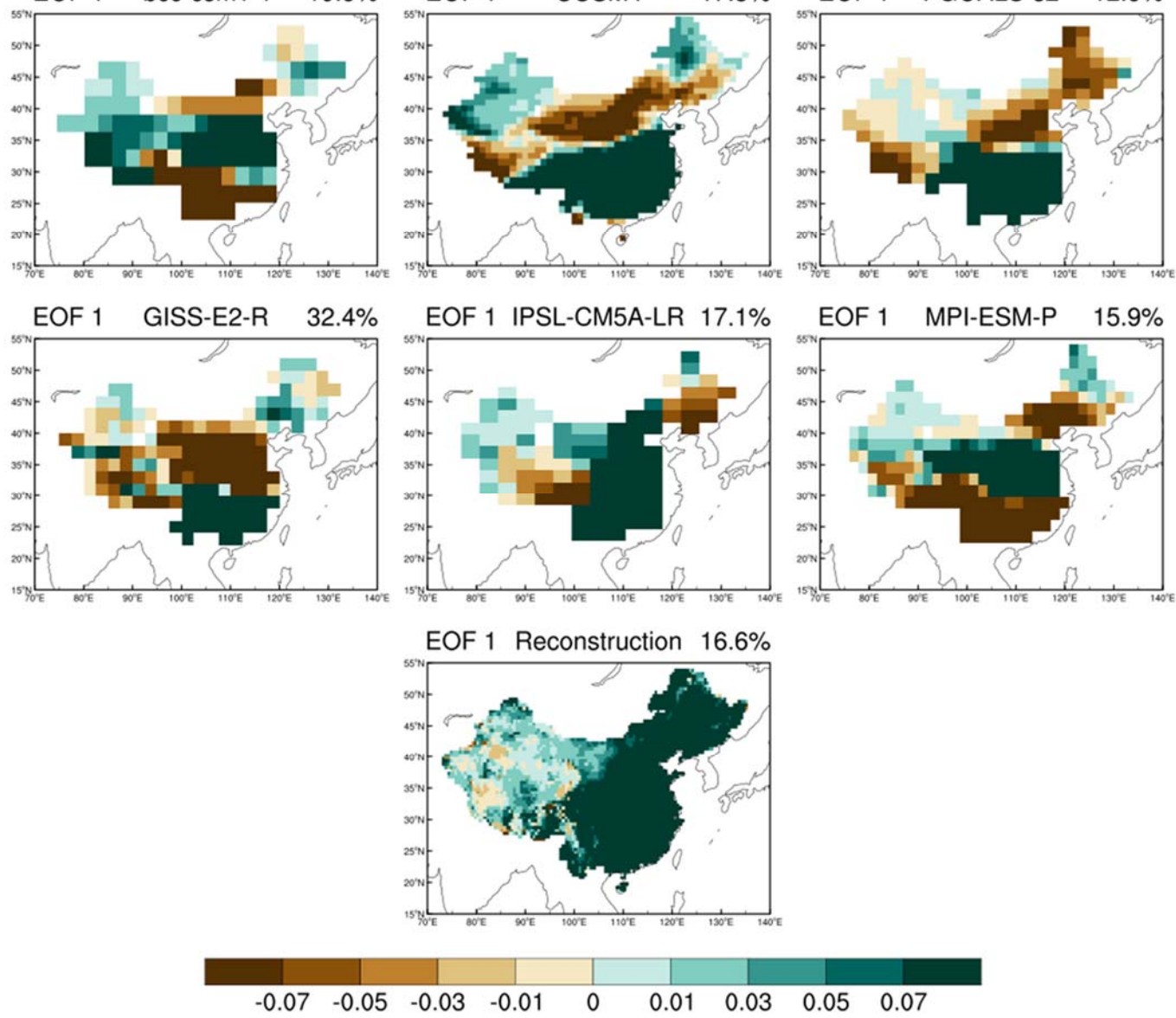

**Figure 8a: Empirical orthogonal function (EOF) 1 of the MJJAS mean precipitation anomalies for China during the interval AD 1470–1849.**

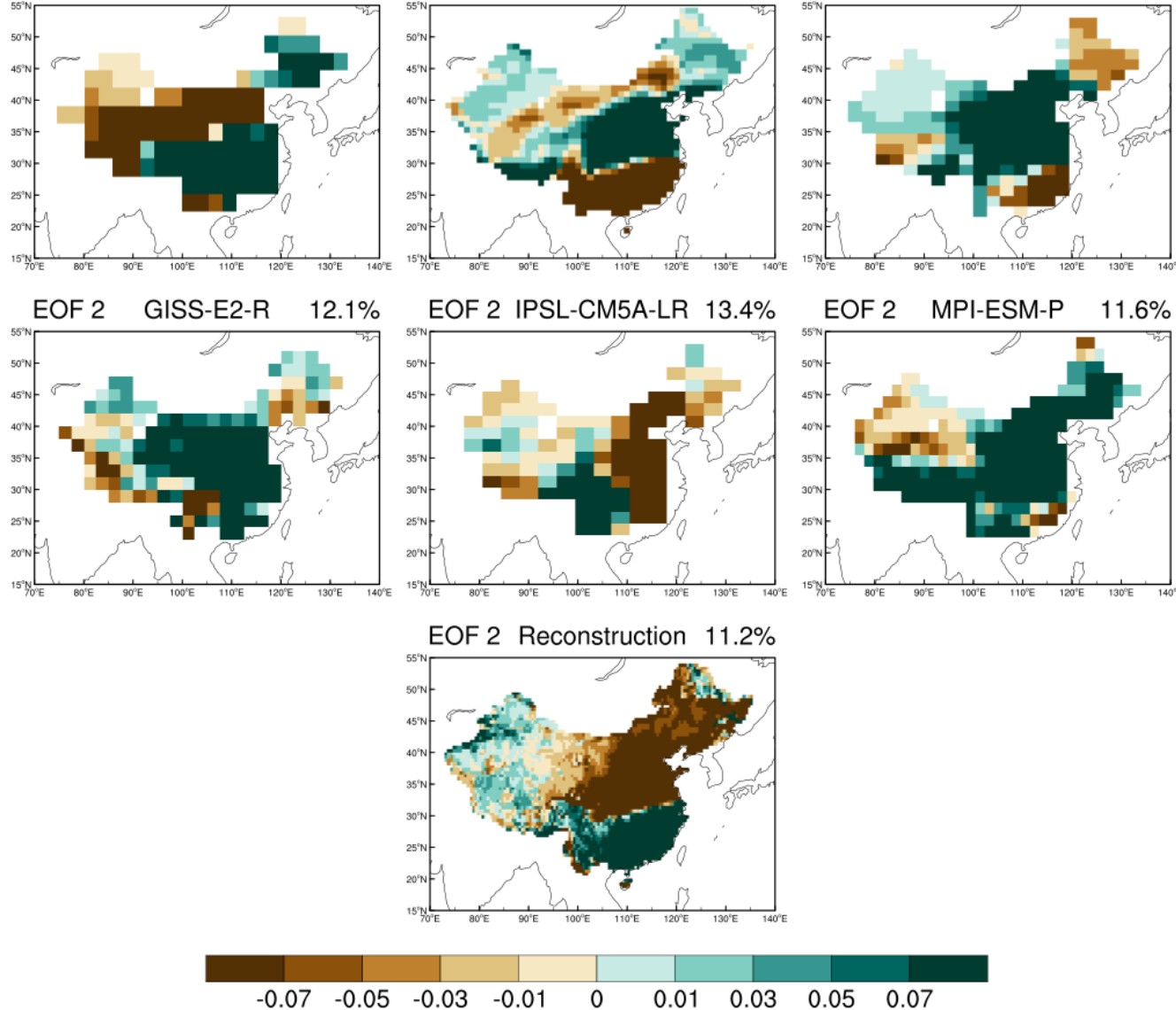

**Figure 8b: Empirical orthogonal function (EOF) 2 of the MJJAS mean precipitation anomalies for China during the interval AD 1470–1849.**

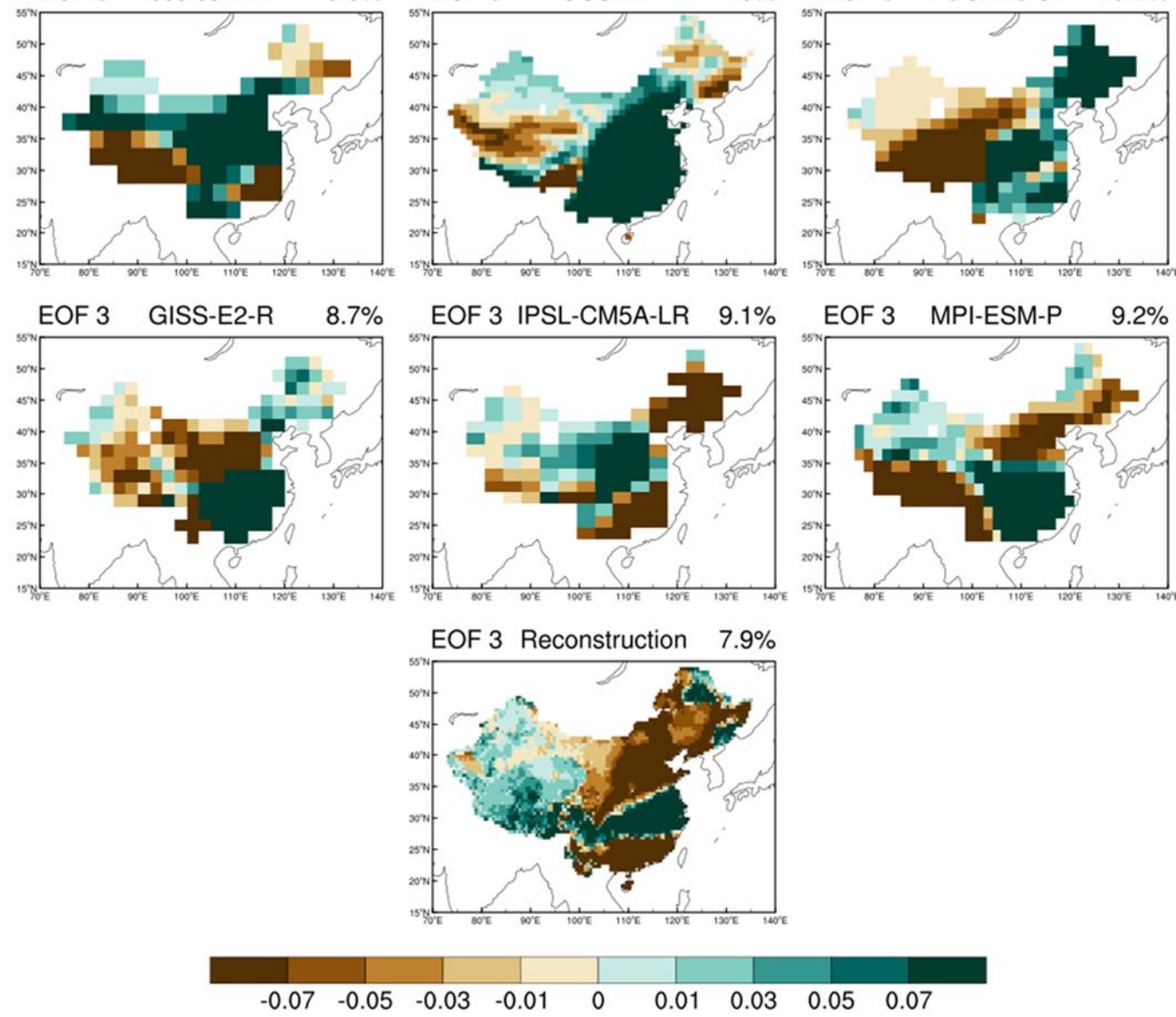

**Figure 8c: Empirical orthogonal function (EOF) 3 of the MJJAS mean precipitation anomalies for China during the interval AD 1470–1849.**

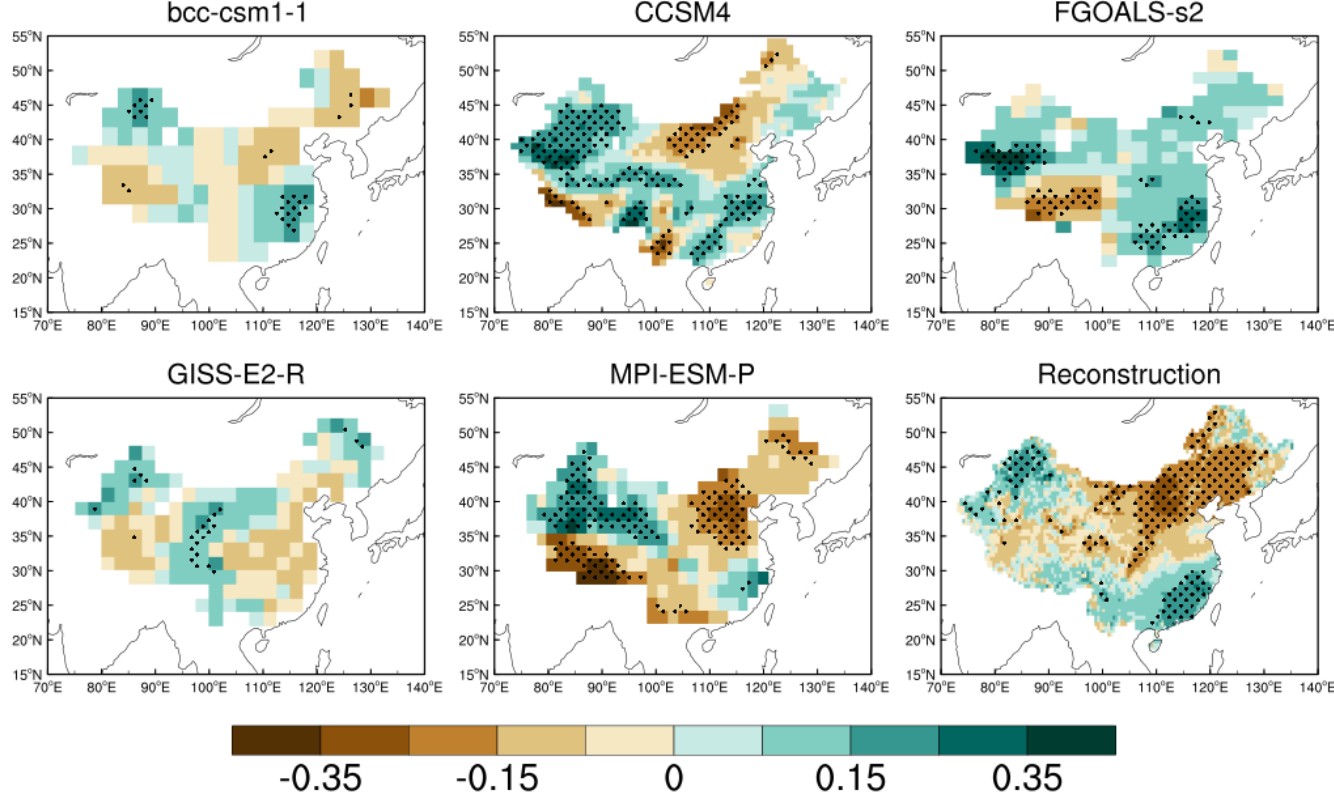

**Figure 9. Correlation maps of five simulated and a reconstructed MJJAS mean precipitation anomalies for China with the five-corresponding simulated annual mean Niño 3.4 indices and a reconstructed annual mean ENSO index (McGregor et al., 2010).**