# Peer review of "Multi-proxy reconstructions of May–September precipitation field in China over the past 500 years"

_Climate of the Past, 2017_

## Referee Comment (RC1) · Anonymous Referee #1 · 26 Feb 2017

The manuscript presents a reconstruction of the seasonal mean precipitation in China over the past few centuries based on a collection of proxy records, comprising mostly tree-ring width and historical documents , but also included some oxygen isotopes record and a long Korean precipitation instrumental record. The statistical method to reconstruct the spatially resolved precipitation is a variant of Point-by-point regression, a method that has been applied for the generation of the Drought Atlases in other continental regions of the world by E. Cook and collaborators. I assume that other reviewers will comment on the quality and adequacy of the proxy records. Here I will mostly focus on the other aspects of the manuscript, like the statistical method applied, the interpretation of the results - connection of the reconstructed precipitation to large-scale variability modes, and on the clarity of the manuscript itself.

In general terms, I think this is a valuable study. The main conclusions related to the

past spatial structure of the precipitation variability, indicating the presence of spatial dipoles at decadal timescales, and the lack of a clear connection to the external forcing are interesting, although maybe to some extend to be expected, and some were already hinted at in previous studies. However, I think the manuscript itself requires some technical revisions, nt dramatic, but indeed careful. The language is sometimes not specific enough and could be misinterpreted by some readers. Also the structure of one section - the discussion-is strange. This section actually contains further results and not so much a discussion about the results. All in all, I would recommend the publication after some revisions, as specified below. Some of my points are related to language usage, but those are more recommendations to check, as I am not a native English speaker

1. The title could be more specific. The study reconstruct seasonal mean precipitation, so it should indicate the season

2. relationships with instrumental climate data (Fritts, 1976; Zhang, 1991). Other proxy records (e.g., ice core, coral, and varve sediment) have been introduced into regional climate field reconstructions (e.g., Neukom et al., 2011), but they are generally harder to use.

This is an example of what I meant by unspecific language, which can be also seen in other parts of the manuscript. Wha does 'harder to use' mean ? I guess the difficulties are related to dating and time resolution, but the authors could be more specific and do not leave the reader guessing.

3. The targets for reconstructions are primarily on temperature variables or variables related to temperature b

The targets are temperature or temperature-related variables.

4. large spatial coherency. Reconstructions of the localized precipitation field or other variables related to precipitation are seldom (Cook et al., 2004; Cook et al., 2015b;

Seftigen et al., 2015) because they require proxy records with more extensive distributions. In particular, the Palmer Drought Severity Index (PDSI) Atlases over the past millennium in North America

I think the authors do not mean more extensive spatial distribution, but rather a more dense proxy network that it would be the case for temperature.

5. The climate field reconstruction method can be divided into the Empirical Orthogonal Function-based (EOF-based) method (Mann et al., 2009) and the point-to-point regression-based (PPR-based) method (Cook et al., 1999). The core function of the

I had real problems with this sentence. I think I understand what the authors mean, but the sentence can be really misleading. First, there are more 'families' of reconstruction methods - consider for instance the Bayesian Hierarchical Modelling Barcast, or the methods based on Canonical Correlation, or the more modern methods based on off-line data assimilation (e.g. Steiger and Hakim) or even the method based on particle filters . Also, the RegEM method used by Mann et al is not really 'EOF-based. It is correct that Mann et al used an EOF pre-filtering within the RegEM method, but this is not required by the algorithm itself. Therefore, I do not think that tis sentence is really correct. The authors may want to re-consider according with what I think they really want to say. They probably mean that statistical methods may include an EOF-prefiltering of the predictand or of the predictor or of both, or not pre-filtering at all. In the former case some small-scale information is lost - I think this is what the authors are pointing to.

6. The left EOF patterns may retain some useful regional spatial information, which would have been partially lost in the EOF- based method. For instance, the global temperature field reconstruction using the EOF-based method (Mann et al., 2009) was

The 'left patterns' is unfortunate. It my be be misinterpreted as 'left and right vectors' in SVD. I would rather used 'the discarded EOF patterns after EOF-truncation'

7. (Shi et al., 2015a) and the optimal information extraction (OIE) method (Yang et al., 2016). In theory, the PPR-based method maximizes the retention of spatial information, but this method requires a sufficient number of suitable proxy

I also had problems with the description of the OIE method, and also to figure out to what extent this method is different from the PPR method. This manuscript does not give enough details and refers to other previous manuscript by Shi et al. I have quickly looked into those papers and I cannot tell the difference between OIE and PPR. This may be my probable, or the problem in previous manuscripts, but I really would recommend to be much more specific here, and at least indicate the basic difference between OIE and PPR, and what are the advantages, if any, of OIE over PPR in this setting.

8 The precipitation (or the variable sensitive to precipitation) field reconstruction for a large-scale region using the PPR-based method is difficult when only one type of proxy records did not cover all reconstruction areas. For example, the tree-ring

This sentence is too cumbersome. I tink I understand what it means, but the authors may consider rephrasing.

9. regression and inverse regression. The LOC regression method has already been verified to efficiently retain low-frequency climate signals (Christiansen, 2011; Shi et al., 2012).

However, the LOC method has been shown to potentially overestimate the past variability. There is a comment and reply exchange on the Christiansen et al manuscript, and my interpretation of it is that Christian et al. also acknowledge that this could be a problem in certain circumstances.

10 2.2 Tree-ring record

Please, be more specific here : three-ring width, isotopes, density, eraly woood density, etc.

11 To maximize the overlap lengths of the instrument data and proxy records, all tree-ring records were extrapolated to AD 2000 using the RegEM algorithm (Schneider, 2001). Here, the truncation parameters for the RegEM algorithm were set to

Extrapolation does not include new information and therefore it cannot increase the skill of the reconstructions. Was this step necessary for the OIE algorithm ? if not, an explanation is required as to why the records were extrapolated.

12 Discussion section. As I indicated in the preamble, this section actually contains further results, such as the superposed epoch analysis. It also contains the analysis of the link between the reconstructed precipitation and ENSO and the PDO. As it stands, it is a classical results section. The title' discussion' is misleading .

13. The superposed epoch analysis (SEA) between the precipitation, its PC1, and 35 large eruption events during AD 1470-1849 shows that volcanic activity as one important external forcing may affect the MJJAS precipitation anomalies variability for China (Fig. 8). Nevertheless, the signals are barely significant and there are similar averaged scores before and after the

These results are too cryptic. The SEA has not been mentioned before, so the reader is left wondering where this comes from: which eruptions have been included, how were they dated (the reconstructed volcanic forcing of Gao et al and of Crowley and Untermann doe not always agree on the dating of the forcing maximum), how was the SEA itself conducted, for instance how many years prior to the eruptions were considered to define the pre-eruption mean, how was the statistical significance established, etc.

14.

Our results indicate thus that the south-north mode variability of precipitation anomalies in China carries very likely the fingerprint of ENSO evolution over the past 500 years, but the origin of the EOF1 and EOF3 patterns are not clearly established yet. This

implies that the other factors such as North Atlantic Oscillation (NAO) (Wu et al., 2009; Zheng et al., 2016), interdecadal Pacific oscillation (IPO) (Song and Zhou, 2015), North Atlantic triple SST pattern (Ruan and Li, 2016) through the North Atlantic–Eurasia Teleconnection (AEAT) (Li et al., 2013a), the snow cover change of the Tibetan Plateau (Ding et al., 2009; Wu et al., 2012), and changes aerosol concentration (Li et al., 2016) may contribute to the reconstructed precipitation field modes during the pre-industrial period.

This conclusion is rather speculative. Why shoud EOF1 and EOF3 be related to the large-scale climate ? they coud be originated by regional processes in China.

15. Caption Figure 1. Please indicate what RSQ, RE, CE and uncertainty mean

―――――――――――――――――

---

## Referee Comment (RC2) · Anonymous Referee #2 · 19 Apr 2017

This is an interesting paper which introduces a newly generated annual warm season precipitation reconstruction over China. The reconstruction is based on the point-to-point regression-based method and a dense data network including 489 tree-ring width data, 2 tree-ring isotope data, 108 drought/flood index, and 1 long-term instrumental data. The verification results show good agreements between the reconstruction and instrumental data over eastern China. The paper is in itself interesting, but its structure and language needs to be improved. Since the methodologies have been commented by another reviewer, here I mainly add some comments about the proxy records.

1. "Each record is required to be significantly correlated with one or more instrumental precipitation record at the 90% (p < 0.1) confidence level during the overlap period, based on both raw data and linearly detrended data. "

[Figure]

How did you do the correlation analysis? How many nearby instrumental grid cells did you compare the proxy data with? It is surprising that all 489 tree-ring proxy records are "sensitive precipitation proxy records".

2. The quality of DWI before the instrumental period needs to be discussed.

3. As you mentioned the precipitation/ PDSI reconstructions over East Asia from Cook et al. (2010) and Feng et al. (2013), have you compared this reconstruction with theirs?

4. " A total of 242 of 491 tree-ring chronologies were extrapolated. The maximum and mean extrapolation lengths of the 242 chronologies were 24 years and 10.5 years, respectively. The extrapolation bias was ignored because of the short extrapolation length."

As you mentioned "all tree-ring records were extrapolated to AD 2000", that means lots of infilling data are between 1981-2000CE, which overlaps with the calibration period (1981-2000CE). Have you considered using a longer calibration period and leave out the last ten years?

5. The second and third leading EOF patterns, the south-north dipole and the sandwich triple pattern, are indeed found in instrumental period. And they are closely related to the movement and intensity of the western pacific subtropical high (WPSH). The sea surface temperature can influence the rainfall through WPSH, but correlation maps between ENSO and summer precipitation are often noisy (Wu and Wang 2002). So, why would the signal of ENSO in rainfall be so strong in the reconstructions if it is noisy in instrumental rainfall data?

6. In Figure 9, which method did you use for the correlation analysis? How does it look using winter temperature in the Nino 3.4 region?

Reference:

Wu, R., & Wang, B. (2002). A contrast of the East Asian summer monsoon–ENSO relationship between 1962–77 and 1978–93. Journal of Climate, 15(22), 3266-3279.

---

## Short Comment (SC1) · 3 May 2017

The PAGES Data Stewardship Integrative Activity seeks to advance best practices for sharing data generated and assembled as part of all PAGES-related activities. As part of this activity, a team of reviewers has been constituted for the "Climate of the Past 2000 years" Special Issue. The data team is reviewing the data handling within each of the CP-Discussion papers in relation to the CP data policy and current best practices. The team has identified essential and recommended additions for each paper, with the goal of achieving a high and consistent level of data stewardship across the 2k Special Issue. We recognize that an additional effort will likely be required to meet the high level of data stewardship envisaged, and we appreciate dedication and contribution of the authors. This includes the use of Data Citations (see example in supplement). We ask authors to respond to our comments as part of the regular open interactive discussion.

[Figure]

If you have any questions about PAGES Data Stewardship principles, please contact any of us directly.

Best wishes for the success of your paper,

2k Special Issue Data Review Team (Darrell Kaufman, Nerilie Abram, Belen Martrat, Raphael Neukom, Scott St. George) and ex-officio team members (Marie-France Loutre, Lucien von Gunten)

Essential additions for this paper:

(1) Add a "data availability" section that describes where the data can be accessed, including a Data Citation for the precipitation reconstruction generated in this study (#6 below).

(2) Add Data Citations (in addition to publication citations) for all 492 proxy datasets listed in Table S1.

(3) For those data not already in a public repository, including the 108 time series of wet/dry records, submit essential metadata along with the time series and include the Data Citation in Table S1.

(4) For those records with previous PAGES 2k IDs, please include cross references to those IDs. These can be found in Table 1 of PAGES2k Consortium (in press). It will be of interest to know how many of the tree rings were used to reconstruct temperature.

(5) If this study relied on the the PAGES2k temperature data base for metadata, quality control or other aspects, then please cite that compilation (PAGES2k Consortium, in press).

(6) Submit the primary outcome of the data analyses to a public repository and include the Data Citation. This includes (a) the detrended and infilled version of all tree-ring chronologies, (b) the resulting precipitation time series reconstruction with and without 9-year smoothing, (c) the IMFs of the reconstruction time series at multiple temporal

scales (Figs 5 and S1), and (d) the reconstructed precipitation value for each grid point at a reasonable time resolution, possibly decadal.

Recommended elements are:

(1) Archive the instrumental precipitation target time series (Fig 4a and 4b, black lines)

(2) Archive the precipitation anomalies for the six climate modes (Fig 9, blue lines)

(3) Archive the code used to generate the reconstruction

Please also note the supplement to this comment:
http://www.clim-past-discuss.net/cp-2017-2/cp-2017-2-SC1-supplement.pdf

**Supplement:**

Data Citations track the provenance of a dataset giving credit to the data generator; this is in addition to any references to publications where the data are described. Data Citations are used in the text (or tables) alongside and in the same way as publication citations. In the Reference list, they include: Creators, Title, Repository, Identifier, Submission Year. More information about Data Citations is here: <https://www.datacite.org/mission.html>

Here is an example of text and corresponding citations (using CP punctuation style):

The PAGES2k Consortium (in press) assembled a large global dataset of temperature-sensitive proxy records (PAGES2k Consortium, 2017). Among the records is the paleo-temperature reconstruction from Laguna Chepical (de Jong et al., 2016), which was described by de Jong et al. (2013).

de Jong, R., von Gunten, l., Maldonado, A., and Grosjean, M.: Late Holocene summer temperatures in the central Andes reconstructed from the sediments of high-elevation Laguna Chepical, Chile (32° S), Climate of the Past, 9, 1921-1932, 2013.

de Jong, R., von Gunten, l., Maldonado, A., and Grosjean, M.: Laguna Chepical summer temperature reconstruction, World Data Center for Paleoclimatology, https://www.ncdc.noaa.gov/paleo/study/20366, 2016.

PAGES 2k Consortium: A global multiproxy database for temperature reconstructions of the Common Era, Scientific Data, in press.

PAGES 2k Consortium: A global multiproxy database for temperature reconstructions of the Common Era, version 2.0.0, figshare, https://figshare.com/s/d327a0367bb908a4c4f2, 2017.

---

## Author Comment (AC2) · 1 Jun 2017

**Interactive comment on "Multi-proxy reconstructions of May–September precipitation field in China over the past 500 years"**

Feng Shi[1,2*], Sen Zhao[3,4], Zhengtang Guo[1,5,6], Hugues Goosse[2], Qiuzhen Yin[2]

[1]Key Laboratory of Cenozoic Geology and Environment, Institute of Geology and Geophysics, Chinese Academy of Sciences, Beijing, 100029, China

[2]Georges Lemaître Centre for Earth and Climate Research, Earth and Life Institute, Université catholique de Louvain, Louvain-la-Neuve, 1348, Belgium

[3]Key Laboratory of Meteorological Disaster of Ministry of Education, and College of Atmospheric Science, Nanjing University of Information Science and Technology, Nanjing, 210044, China

[4]School of Ocean and Earth Sciences and Technology, University of Hawaii at Mānoa, Honolulu, HI, 96822, USA

[5]CAS Center for Excellence in Tibetan Plateau Earth Sciences, Beijing, 100101, China

[6]University of Chinese Academy of Sciences, Beijing, 100049, China

*Correspondence to*: Feng SHI (shifeng@mail.iggcas.ac.cn)

- In blue: referees' comments

- In black: our answers

- In black italic: what we will propose to add in the text
* * *
**General comments:**

This is an interesting paper which introduces a newly generated annual warm season precipitation reconstruction over China. The reconstruction is based on the point-to-point regression-based method and a dense data network including 489 tree-ring width data, 2 tree-ring isotope data, 108 drought/flood index, and 1 long-term instrumental data. The verification results show good agreements between the reconstruction and instrumental data over eastern China. The paper is in itself interesting, but its structure and language needs to be improved. Since the methodologies have been commented by another reviewer, here I mainly add some comments about the proxy records.

**Response:** We would like to thank the reviewer for his/her constructive review, comments, and suggestions, which have helped us to greatly improve our manuscript.

We have done our best to address the reviewer's concerns and modified the manuscript in light of the reviewer's suggestions, in particular to improve the structure and the language. Point-by-point responses to the reviewer's comments are listed below.

**Major comments:**

**Item 1:** 1. "Each record is required to be significantly correlated with one or more instrumental precipitation record at the 90% (p < 0.1) confidence level during the overlap period, based on both raw data and linearly detrended data."

How did you do the correlation analysis? How many nearby instrumental grid cells did you compare the proxy data with? It is surprising that all 489 tree-ring proxy records are "sensitive precipitation proxy records ".

**Response:** We calculated the Pearson's linear correlation coefficient between the targeted precipitation grid point and the candidate tree-ring record which is located in the range of the search radius, and select all tree-ring records that are positively and significantly related with one instrumental precipitation grid points during the period AD 1961-2000 at least (r > 0 and p < 0.1). Figure 1 shows the number of grid points that is reconstructed using each tree-ring record. e.g. a tree-ring record with "100" value means that it has significant relationship with 100 nearby instrumental precipitation grid points. This illustrates that all tree-ring records are significantly related to a precipitation grid point at least.

Thanks to your question, we have revised this statement, it is '*proxy records that can be related to precipitation in the domain studied*' not '*sensitive precipitation proxy records*', because we do not determine whether the precipitation is the major limit factor of tree-ring width chronology and use all possible predictors to extend the spatial coverage.

[Figure]

Figure 1 The number of grid points which is reconstructed using each tree-ring record.

**Item 2:** 2. The quality of DWI before the instrumental period needs to be discussed.

**Response:** Following the reviewer's suggestion, we will add two figures and a discussion on the quality of DWI in the revised manuscript. The reliability of DWI was demonstrated in Zhang (1988). Firstly, Figures 2 and 3 both show that the weakness of DWI are time discontinuity and uneven distribution (Zhang, 1988). Figure 2 is the time spans of 107 DWI records. This illustrates that most of DWI records are not continuous in time. The maximum (mean) number of missing values of 107 DWI records during the period AD 1470-2000 was 484 (161). Figure 3 shows that the number of observation of 107 DWI records during the period AD 1470-2000. A DWI record with "100" value means that it has 100 observed values during the period AD 1470-2000. This exhibits that 99 of 107 DWI records located in Eastern China (east of longitude 105°E), which is the economically developed region. There are only six DWI records to the west of longitude 100°E. The numbers of observation of these six DWI records are all less than 140. Additionally, an uncertainty due to the subjective judgment is unavoidable for the historical documentary, even though the different sources have been used to cross-validate the final reconstruction (Gong, et al., 1983; Mou, 1996; Man, 2009; Ge, 2011; Zheng et al., 2014). Moreover, the range of values is within five grades and the DFI record is not an accurate precipitation value, thus it also limits the accuracy and ability to detect the extreme events (Zheng et al., 2014).

[Figure]

Figure 2 The time span of each Dryness/Wetness index.

[Figure]

Figure 3 The number of observation of each Dryness/Wetness index during the period AD 1470-2000.

**Item 3:** 3. As you mentioned the precipitation/ PDSI reconstructions over East Asia from Cook et al. (2010) and Feng et al. (2013), have you compared this reconstruction with theirs?

**Response:** Following the reviewer's suggestion, we have calculated the correlation between Cook's PDSI reconstruction (Cook et al., 2010) with our precipitation reconstruction in Figure 4.

[Figure]

Figure 4 Spatial correlation between the PDSI reconstruction (Cook et al., 2010) and the precipitation field reconstruction obtained in this study. The dark spots mean the significant levels of the correlation coefficients at the 99% confidence level.

Figure 4 shows that the spatial correlation between them. The strong correlations appear in the Northeast Tibetan Plateau, where there are longest and most abundant tree-ring width chronologies in China. Here, the precipitation is possibly a primary control factor of the tree-ring width chronology (Yang et al., 2014b;Zhang et al., 2003). There are weak correlations between the reconstructions in Eastern China even through the correlation coefficients are significant in some regions. The Monsoon Asia Drought Atlas (MADA) is not consistent with the DWI records in Eastern China but only very few and short tree-ring width chronologies in Eastern China are used to reconstruct the MADA (Yang et al., 2013;Kang et al., 2014;Yang et al., 2014a;Zheng et al., 2014;Ge et al., 2016). We will include Figure 4 as supplementary material and discuss it in the revised manuscript.

[Figure]

Figure 5 Spatial patterns of the May–September precipitation field relative to the 1961–90 climatological mean during five severe droughts in China.

It is a pity that the precipitation reconstruction (Feng et al., 2013) is not achieved in any published database. However, a qualitative comparison is possible for the five historical Chinese droughts in Figure 5, which will be included in the revised manuscript. The selection of five drought periods follows Feng et al. (2013). The spatial patterns of five-drought events in Figures 5a−e are different with Feng et al., (2013). A "north drought with south flooding" dipole pattern in Eastern China is seen in Figure 5a−d, and there is a triple pattern in Eastern China in Figure 5f. The similar dipole pattern can be found in Figure 5b and 5d (Feng et al., 2013) for the two drought events (AD 1586−89 and AD 1876−78). but during the above five drought periods, most of northeastern China is relatively humid situation in our study, which is not consistent with Feng et al. (2013)'s Figure 5. Moreover, the reconstruction skills are low in semiarid and arid regions (Feng et al., 2013), because there are very few tree-ring chronologies in Western China in that reconstruction. We used some tree-ring width chronologies in Western China to compensate this weakness.

**Item 4:** 4. "A total of 242 of 491 tree-ring chronologies were extrapolated. The maximum and mean extrapolation lengths of the 242 chronologies were 24 years and 10.5 years, respectively. The extrapolation bias was ignored because of the short extrapolation length."

As you mentioned "all tree-ring records were extrapolated to AD 2000", that means lots of infilling data are between 1981-2000CE, which overlaps with the calibration period (1981-2000CE). Have you considered using a longer calibration period and leave out the last ten years?

**Response:** The extrapolation is necessary for the OIE method, because we used the correlation coefficient between the candidate proxy record and the reconstructed target to weight the candidate proxy records. Following the reviewer's suggestion, we have used a longer calibration period (1961-1990) and the last ten years as the verification period (1991-2000). The corresponding contents will be revised according to the updated results.

**Item 5:** The second and third leading EOF patterns, the south-north dipole and the sandwich triple pattern, are indeed found in instrumental period. And they are closely related to the movement and intensity of the western pacific subtropical high (WPSH). The sea surface temperature can influence the rainfall through WPSH, but correlation maps between ENSO and summer precipitation are often noisy (Wu and Wang 2002). So, why would the signal of ENSO in rainfall be so strong in the reconstructions if it is noisy in instrumental rainfall data?

**Response:** This is indeed an interesting point. We cannot determine the origin of this behavior but two hypotheses can be proposed: 1) The tree-ring width chronologies in Asia are significantly related to the multiple ENSO indices (Shi et al., 2016), however, this relationship is not stable and time-varying in Figure 6 (Shi et al., 2016). We have calculated the running correlation between the 17 reconstructed ENSO indices and the precipitation anomalies averaged over Yangtze River region and Huai River region in

Figures 6a-b. We have followed Wu and Wang (2002) for the region definition. All data were filtered to obtain the interannual components when calculating running correlation coefficients using the Ensemble Empirical mode decomposition (EEMD) method, because the ENSO variability is in the interannual 3–7 year period band (Trenberth, 1997). Results indicate that the relationship between ENSO and summer precipitation in China are not stable in the reconstructions, which is consistent with the instrumental period (Wu and Wang, 2002). This indicates that different factors may affect the spatial pattern of the precipitation field, e.g. the volcanic eruption events and a more robust link is found if longer time series are analyzed. 2) The background climates states between the reconstruction period AD 1470-1849 and the instrumental period AD 1962-1993 are different. The increasing anthropogenic factor may have important influences on the precipitation field in China, e.g. the changes aerosol concentration (Li et al., 2016) and may mask some of the links between ENSO and precipitation over China. This will be discussed more explicitly in the revised version.

[Figure]

Figure 6 The 101-year running correlation coefficients of the interannual components of 17 ENSO indices with the reconstructed precipitation anomalies averaged over Yangtze River region (28°–32°N, 110°–120°E; a) and Huai River region (32.5°–35.5°N, 115°–120°E; b). The interannual components were obtained using the Ensemble Empirical mode decomposition (EEMD) method.

Moreover, we will have added the statement that the two leading EOF patterns are closely related to the western pacific subtropical high (Wu and Wang, 2002) in the revised manuscript.

**Item 6:** 6. In Figure 9, which method did you use for the correlation analysis? How does it look using winter temperature in the Nino 3.4 region?

**Response:** In Figure 9, the Pearson's sample linear cross-correlations at lag 0 is used for the correlation analysis, and the effective number of degrees of freedom is calculated following Zhao et al. (2016) to access the significance of the correlation. The unified ENSO proxy (UEP) (McGregor et al., 2010) is used to calculate the relationship between the precipitation field and the ENSO index. This index is the first PC of ten reconstructed ENSO indices, and is scaled to the instrumental HadSST1 annual mean (July–June) Niño 3.4 region sea surface temperature anomalies (McGregor et al., 2010). We will have added this explanation in the method section. Moreover, we have calculated the relationship between the previous winter (December–January–February) Niño 3.4 index and the precipitation field for each model in Figure 7. The spatial pattern is very similar with the result of annual mean Niño 3.4 index.

[Figure]

Figure 7 (a-e) Correlation maps of simulated MJJAS mean precipitation anomalies for China with the corresponding simulated preceding winter (DJF) Niño 3.4 index in five different models (bcc-csm1-1, CCSM4, FGOALS-s2, GISS-E2-R, and MPI-ESM-P). (f) Correlation maps of reconstructed MJJAS mean precipitation anomalies for China with a reconstructed annual mean ENSO index from McGregor et al. (2010).

References

Cook, E. R., Anchukaitis, K. J., Buckley, B. M., D'Arrigo, R., Jacoby, G. C., and Wright, W. E.: Asian monsoon failure and megadrought during the last millennium, Science, 328, 486-489, 2010.

Feng, S., Hu, Q., Wu, Q., and Mann, M. E.: A gridded reconstruction of warm season precipitation for Asia spanning the past half millennium, J. Climate, 26, 2192-2204, 2013.

Ge, Q.: The climate change in China during the past dynasties, Science Press, Beijing, 2011. (in Chinese)

Ge, Q., Zheng, J., Hao, Z., Liu, Y., and Li, M.: Recent advances on reconstruction of climate and extreme events in China for the past 2000 years, J. Geog. Sci. , 26, 827-854, 2016.

Gong, G., Zhang, P., Wu, X. and Zhang, J.: Research methods of historical climate change, Science Press, Beijing, 1983. (in Chinese)

Kang, S., Bräuning, A., and Ge, H.: Tree-ring based evidence of the multi-decadal climatic oscillation during the past 200 years in north-central China, J. Arid Environ., 110, 53-59, 2014.

Li, Z., Lau, W. K. M., Ramanathan, V., Wu, G., Ding, Y., Manoj, M. G., Liu, J., Qian, Y., Li, J., Zhou, T., Fan, J., Rosenfeld, D., Ming, Y., Wang, Y., Huang, J., Wang, B., Xu, X., Lee, S. S., Cribb, M., Zhang, F., Yang, X., Zhao, C., Takemura, T., Wang, K., Xia, X., Yin, Y., Zhang, H., Guo, J., Zhai, P. M., Sugimoto, N., Babu, S. S., and Brasseur, G. P.: Aerosol and monsoon climate interactions over Asia, Rev. Geophys., 54, 866-929, 2016.

Man, Z.: Climatic change historical period of China, Shandong Education Press, Jinan, 2009. (in Chinese)

McGregor, S., Timmermann, A., and Timm, O.: A unified proxy for ENSO and PDO variability since 1650, Clim. Past, 6, 1-17, 2010.

Mou, C.: Further research on the climatic fluctuations during the last 5000 years in China, China Meteorological Press, Beijing, 1996. (in Chinese)

Shi, F., Fang, K., Xu, C., Guo, Z., and P., B. H.: Interannual to centennial variability of the South Asian summer monsoon over the past millennium, Clim. Dyn., doi:10.1007/s00382-016-3493-9, 2016.

Trenberth, K. E.: The definition of el nino, Bull. Am. Meteorol. Soc., 78, 2771-2777, 1997.

Wang, P., and Zhang, D.: An introduction to some historical governmental weather records of China, Bull. Am. Meteorol. Soc., 69, 753-758, 1988.

Wu, R., and Wang, B.: A contrast of the East Asian summer monsoon–ENSO relationship between 1962–77 and 1978–93, J. Climate, 15, 3266-3279, 2002.

Yang, B., Kang, S., Ljungqvist, F. C., He, M., Zhao, Y., and Qin, C.: Drought variability at the northern fringe of the Asian summer monsoon region over the past millennia, Clim. Dyn., 43, 845-859, 2014a.

Yang, B., Qin, C., Wang, J., He, M., Melvin, T. M., Osborn, T. J., and Briffa, K. R.: A 3,500-year tree-ring record of annual precipitation on the northeastern Tibetan Plateau, Proc. Nat. Acad. Sci. U.S.A., 111, 2903-2908, 2014b.

Yang, F., Shi, F., Kang, S., Wang, S., Xiao, Z., Nakatsuka, T., and Shi, J.: Comparison of the dryness/wetness index in China with the Monsoon Asia Drought Atlas, Theor. Appl. Climatol., 114, 553-566, 2013.

Zhang, D.: The method for reconstruction of the dryness/wetness series in China for the last 500 years and its reliability, In: Zhang, J. (ed): The reconstruction of climate in China for historical times. Science Press, Beijing, 1988.

Zhang, Q., Cheng, G., Yao, T., Kang, X., and Huang, J.: A 2,326-year tree-ring record of climate variability on the northeastern Qinghai-Tibetan Plateau, Geophys. Res. Lett., 30, 1739-1742, 2003.

Zhao, S., Li, J., and Sun, C.: Decadal variability in the occurrence of wintertime haze in central eastern China tied to the Pacific Decadal Oscillation, Sci. Rep., 6, doi:10.1038/srep27424, 2016.

Zheng, J., Ge, Q., Hao, Z., Liu, H., Man, Z., Hou, Y., and Fang, X.: Paleoclimatology proxy recorded in historical documents and method for reconstruction on climate change, Quat. Sci., 34, 1186-1196, 2014. (in Chinese with English abstract)

Zheng, J., Hao, Z., Fang, X., and Ge, Q.: Changing characteristics of extreme climate events during past 2000 years in China, Prog. Geogr., 33, 3-12, 2014. (in Chinese with English abstract)

Zhang, P.: Climate change in China during historical times, Shandong Science and Technology Press, Jinan, 1996. (in Chinese)

---

## Author Comment (AC3) · 1 Jun 2017

**Interactive comment on "Multi-proxy reconstructions of May–September precipitation field in China over the past 500 years"**

Feng Shi[1,2*], Sen Zhao[3,4], Zhengtang Guo[1,5,6], Hugues Goosse[2], Qiuzhen Yin[2]

[1]Key Laboratory of Cenozoic Geology and Environment, Institute of Geology and Geophysics, Chinese Academy of Sciences, Beijing, 100029, China

[2]Georges Lemaître Centre for Earth and Climate Research, Earth and Life Institute, Université catholique de Louvain, Louvain-la-Neuve, 1348, Belgium

[3]Key Laboratory of Meteorological Disaster of Ministry of Education, and College of Atmospheric Science, Nanjing University of Information Science and Technology, Nanjing, 210044, China

[4]School of Ocean and Earth Sciences and Technology, University of Hawaii at Mānoa, Honolulu, HI, 96822, USA

[5]CAS Center for Excellence in Tibetan Plateau Earth Sciences, Beijing, 100101, China

[6]University of Chinese Academy of Sciences, Beijing, 100049, China

*Correspondence to*: Feng SHI (shifeng@mail.iggcas.ac.cn)

- In blue: referees' comments

- In black: our answers

- In black italic: what we will propose to add in the text

————————————————————————————————————————————————————

**General comments:**

The PAGES Data Stewardship Integrative Activity seeks to advance best practices for sharing data generated and assembled as part of all PAGES-related activities. As part of this activity, a team of reviewers has been constituted for the "Climate of the Past 2000 years" Special Issue. The data team is reviewing the data handling within each of the CP-Discussion papers in relation to the CP data policy and current best practices. The team has identified essential and recommended additions for each paper, with the goal of achieving a high and consistent level of data stewardship across the 2k Special Issue. We recognize that an additional effort will likely be required to meet the high level of data stewardship envisaged, and we appreciate dedication and contribution of the authors. This includes the use of Data Citations (see example in

supplement). We ask authors to respond to our comments as part of the regular open interactive discussion. If you have any questions about PAGES Data Stewardship principles, please contact any of us directly.

Best wishes for the success of your paper,

2k Special Issue Data Review Team (Darrell Kaufman, Nerilie Abram, Belen Martrat, Raphael Neukom, Scott St. George) and ex-officio team members (Marie-France Loutre, Lucien von Gunten)

**Major comments:**

**Item 1:** Essential additions for this paper:

(1) Add a "data availability" section that describes where the data can be accessed, including a Data Citation for the precipitation reconstruction generated in this study (#6 below).

**Response:** We will add 'data availability' section in the revised manuscript. Our reconstructed precipitation field will be archived on the NOAA website, as done for our previous studies (e.g., Shi et al., 2012a;Shi et al., 2012b;Shi et al., 2013;Shi et al., 2014;Shi et al., 2015). A data citation will be added for all the records. It's a good opportunity to check all datasets again, especially for the proxy records. We found that 31 tree-ring records in the original Table S1 from different sources are duplicated and should be removed. e.g. No. 299 and No. 482 are the same tree-ring width chronology in Zhangxian, China. Moreover, a Dryness/Wetness record in Tainan was not included in any grid reconstruction and can also be deleted. Herein, 476 proxy records are already available in the public repository, including 241 tree-ring width chronologies in International Tree Ring Data Bank (ITRDB) (https://www.ncdc.noaa. gov/data-access/paleoclimatology-data/datasets/tree-ring), 78 tree-ring width chronologies in the supplement of the book (Li et al., 2000), 38 tree-ring chronologies in the PAGES2k_dataset v2.0 (https://figshare.com/s/d327a0367bb908a4c4f2), 12 tree-ring width chronologies and 107 Dryness/Wetness records at the webpages of the Chinese Meteorological Data Service Center (CMDC) (http://data.cma.cn/data/detail/ dataCode/HPXY_HDOC_CHN_DAW.html, and http://data.cma.cn/data/cdcdetail/ dataCode/HPXY_TRRI_CHN.html). Following the PAGES Data Stewardship Integrative Activity, only these 476 proxies that are already archived in public repository will be used to reconstruct the precipitation field in the revised version of this study. All figures will be updated with this dataset. We compared the revised figures with the previous versions, and confirmed that the revision has no substantial impact on our main conclusions.

We have the data analysis to confirm the quality of our reconstruction is similar with only the archived data sets. According to data availability, we divided into three proxy subsets: 1) version A: 476 proxies are already archived in published repository. 2) version B: 38 tree-ring chronologies are included in addition to 476 proxies in

version A. The 38 proxies were shared from Asia2K phase one. We will contact with the leader of Asia2K phase one and hope that the 38 tree-ring chronologies can be archived in published repository soon; 3) version C: 53 tree-ring chronologies and an instrumental record are added in addition for 514 proxies in version B. We do not expect that those 54 records will be publicly available soon. The figures below show differences between the 3 subsets.

**Data coverage**

In Figure 1, the coverages of three subsets are different. For instance, in version C, there is a tree-ring chronology in Northeastern China and a long instrumental precipitation record in South Korea, which are useful to improve the accuracy of reconstruction in Northeastern China.

[Figure]

Figure 1 Map showing the locations of 476 proxy records (a), 514 proxy records (b), and 568 proxy records (c).

**Skill scores of three reconstructions using the 3 data subsets**

Figure 2 shows that the $r^2$, RE and CE values of three reconstruction are very similar. But the four grids ([34.25°N, 100.25°E], [34.25°N, 100.75°E], [41.25°N, 128.25°E], and [49.25°N, 129.25°E]) in version A and the two grids ([34.25°N, 100.25°E] and [34.25°N, 100.75°E]) in version B were not reconstructed because there are no enough proxies.

a) Version A:

[Figure]

b) Version B:

[Figure]

c) Version C

[Figure]

Figure 2 Skills of the reconstructed MJJAS mean precipitation anomalies in China for three versions (the 1961–1990 verification period and the 1991–2000 calibration period). The $r^2$ is the square of the Pearson product–moment correlation coefficient, the RE and CE are the reduction of error and the coefficient of efficiency, the uncertainty is characterized from the standard deviation of the residual between the reconstructed and instrumental precipitation data during the verification period.

**Conclusion:** The skill scores of three reconstructions are very similar. This indicates that the selection of subsets does not affect the data analysis and the main conclusion in this manuscript. But it still affects the several grid reconstructions. e.g. the four grids in version A were not reconstructed because of the insufficient proxy records. Following the rule of the PAGES2k special issue, we will show the results with version A in the revised manuscript.

**Item 2:** (2) Add Data Citations (in addition to publication citations) for all 492 proxy datasets listed in Table S1.

**Response:** Following the data review team's suggestion, we will add the data citations in Table S1 in the revised manuscript.

**Item 3:** (3) For those data not already in a public repository, including the 108-time

series of wet/dry records, submit essential metadata along with the time series and include the Data Citation in Table S1.

**Response:** Following the team's suggestion, we will submit essential metadata which includes the Data Citation in Table S1, but the time series will not be included in Table S1. Because the Dryness/Wetness dataset is already achieved in Chinese Meteorological Data Service Center (CMDC) in a public repository. It is easy to get the time series through a simple registration on the website (http://data.cma.cn/data/detail/dataCode/HPXY_HDOC_CHN_DAW.html).

**Item 4:** (4) For those records with previous PAGES 2k IDs, please include cross references to those IDs. These can be found in Table 1 of PAGES2k Consortium (in press). It will be of interest to know how many of the tree rings were used to reconstruct temperature.

**Response:** Following the team's suggestion, we have checked all Asian tree-ring chronologies in PAGES2k Consortium dataset v2.0 (PAGES2k_v2.0.0). There are 229 Asian tree-ring chronologies in the PAGES2k_v2.0.0 dataset, and 172 of them are used in this study. Cross references will be made in Table S1.

**Item 5:** (5) If this study relied on the PAGES2k temperature data base for metadata, quality control or other aspects, then please cite that compilation (PAGES2k Consortium, in press).

**Response:** Following the team's suggestion, we will cite the compilation (PAGES2k Consortium, in press) for 172 tree-ring chronologies in this study. Herein, 134 of 172 tree-ring chronologies have already been archived on NOAA website.

**Item 6:** (6) Submit the primary outcome of the data analyses to a public repository and include the Data Citation. This includes (a) the detrended and infilled version of all tree-ring chronologies, (b) the resulting precipitation time series reconstruction with and without 9-year smoothing, (c) the IMFs of the reconstruction time series at multiple temporal scales (Figs 5 and S1), and (d) the reconstructed precipitation value for each grid point at a reasonable time resolution, possibly decadal.

**Response:** Following the PAGES Data Stewardship Integrative Activity, we only use 476 proxy records which are already archived in the public repository. We will submit the metadata with the data citation in Table S1. The output of the data analyses will be submitted to NOAA website and a data citation will be added. (a) All 369 tree-ring chronologies are already archived in the public repository. We will include the original data URL to show where the data are archived by the original author. (b) The gridded precipitation data including the regional mean curve (in Figs 4 and 6) and its IMFs (in Figs 5 and S1) will be submitted at an annual time resolution to allow analyzing extreme events.

**Item 7:** Recommended elements are:

(1) Archive the instrumental precipitation target time series (Fig 4a and 4b, black

**Response:** The two instrumental precipitation datasets are both archived by the Chinese Meteorological Data Service Center (CMDC). Herein, the China's Ground Precipitation Dataset V2.0 can be obtained from the website (http://data.cma.cn/data/detail/dataCode/SURF_CLI_CHN_PRE_MON_GRID_0.5.html) and The Homogenized Monthly Precipitation Dataset in China can be downloaded from the website (http://data.cma.cn/data/detail/dataCode/SEVP_CLI_CHN_PRE_MON_GRID.html).

**Item 8:** (2) Archive the precipitation anomalies for the six climate modes (Fig 9, blue lines)

**Response:** We will archive the precipitation anomalies for the five climate models in Fig 9 on NOAA website, the last one is the reconstructed result, not the simulation.

**Item 9:** (3) Archive the code used to generate the reconstruction

**Response:** We will archive the code for the OIE method on GitHub website after this work published.

References

Li, J., Yuan, Y., and You, X.: The tree-ring hydrology research and application, Science Press, Beijing, 2000. (in Chinese)

PAGES 2k Consortium: A global multiproxy database for temperature reconstructions of the Common Era, Sci. Data, in press, 2017.

Shi, F., Yang, B., Charpentier Ljungqvist, F., and Fengmei, Y.: Multi-proxy reconstruction of Arctic summer temperatures over the past 1400 years, Clim. Res., 54, 113-128, 2012a.

Shi, F., Yang, B., and von Gunten, L.: Preliminary multiproxy surface air temperature field reconstruction for China over the past millennium, Sci. China Earth Sci., 55, 2058-2067, 2012b.

Shi, F., Yang, B., Mairesse, A., von Gunten, L., Li, J., Bräuning, A., Yang, F., and Xiao, X.: Northern Hemisphere temperature reconstruction during the last millennium using multiple annual proxies, Clim. Res., 56, 231-244, 2013.

Shi, F., Li, J., and Wilson, R. J. S.: A tree-ring reconstruction of the South Asian summer monsoon index over the past millennium, Sci. Rep., 4, doi:10.1038/srep06739, 2014.

Shi, F., Ge, Q., Yang, B., Li, J., Yang, F., Ljungqvist, F. C., Solomina, O., Nakatsuka, T., Wang, N., Zhao, S., Xu, C., Fang, K., Sano, M., Chu, G., Fan, Z., Narayan, P. G., and Muhammad, U. Z.: A multi-proxy reconstruction of spatial and temporal variations in Asian summer temperatures over the last millennium, Clim. Chang., 131, 663-676, 2015.

---

## Author Response (AR1)

**Interactive comment on "Multi-proxy reconstructions of May–September precipitation field in China over the past 500 years"**

Feng Shi[1,2*], Sen Zhao[3,4], Zhengtang Guo[1,5,6], Hugues Goosse[2], Qiuzhen Yin[2]

[1]Key Laboratory of Cenozoic Geology and Environment, Institute of Geology and Geophysics, Chinese Academy of Sciences, Beijing, 100029, China

[2]Georges Lemaître Centre for Earth and Climate Research, Earth and Life Institute, Université catholique de Louvain, Louvain-la-Neuve, 1348, Belgium

[3]Key Laboratory of Meteorological Disaster of Ministry of Education, and College of Atmospheric Science, Nanjing University of Information Science and Technology, Nanjing, 210044, China

[4]School of Ocean and Earth Sciences and Technology, University of Hawaii at Mānoa, Honolulu, HI, 96822, USA

[5]CAS Center for Excellence in Tibetan Plateau Earth Sciences, Beijing, 100101, China

[6]University of Chinese Academy of Sciences, Beijing, 100049, China

*Correspondence to*: Feng SHI (shifeng@mail.iggcas.ac.cn)

- In blue: referees' comments

- In black: our answers

- In black italic: what we have added in the text
* * *
**I.   Response to reviewer 1:**

**General comments:**

The manuscript presents a reconstruction of the seasonal mean precipitation in China over the past few centuries based on a collection of proxy records, comprising mostly tree-ring width and historical documents, but also included some oxygen isotopes record and a long Korean precipitation instrumental record. The statistical method to reconstruct the spatially resolved precipitation is a variant of Point-by-point regression, a method that has been applied for the generation of the Drought Atlases in other continental regions of the world by E. Cook and collaborators. I assume that other reviewers will comment on the quality and adequacy of the proxy records. Here I will mostly focus on the other aspects of the manuscript, like the statistical method applied,

the interpretation of the results - connection of the reconstructed precipitation to largescale variability modes, and on the clarity of the manuscript itself.

In general terms, I think this is a valuable study. The main conclusions related to the past spatial structure of the precipitation variability, indicating the presence of spatial dipoles at decadal timescales, and the lack of a clear connection to the external forcing are interesting, although maybe to some extend to be expected, and some were already hinted at in previous studies. However, I think the manuscript itself requires some technical revisions, not dramatic, but indeed careful. The language is sometimes not specific enough and could be misinterpreted by some readers. Also, the structure of one section - the discussion-is strange. This section actually contains further results and not so much a discussion about the results. All in all, I would recommend the publication after some revisions, as specified below. Some of my points are related to language usage, but those are more recommendations to check, as I am not a native English speaker

**Response:** We would like to thank the reviewer for his/her constructive review, comments, and suggestions, which have helped us to greatly improve our manuscript. We have done our best to address the reviewer's concerns and modified the manuscript in light of the reviewer's suggestions. In particular, we have modified the structure and the discussion following the reviewer's advice. Point-by-point responses to the comments are listed below.

**Major comments:**

**Item 1:** 1. The title could be more specific. The study reconstructs seasonal mean precipitation, so it should indicate the season

**Response:** The title has been revised to "*Multi-proxy reconstructions of May–September precipitation field in China over the past 500 years*" according to the reviewer's suggestion.

**Item 2:** 2. relationships with instrumental climate data (Fritts, 1976; Zhang, 1991). Other proxy records (e.g., ice core, coral, and varve sediment) have been introduced into regional climate field reconstructions (e.g., Neukom et al., 2011), but they are generally harder to use.

This is an example of what I meant by unspecific language, which can be also seen in other parts of the manuscript. What does 'harder to use' mean? I guess the difficulties are related to dating and time resolution, but the authors could be more specific and do not leave the reader guessing.

**Response:** The sentence has been revised to "*they generally represent a small percentage of the data available in global compilations (e.g. PAGES-2k-Consortium, 2013).*", according to the reviewer's suggestion. Moreover, the other parts of the manuscript have also checked and polished.

**Item 3:** 3. The targets for reconstructions are primarily on temperature variables or

variables related to temperature b

The targets are temperature or temperature-related variables.

**Response:** The sentence has been revised to "*The reconstruction targets are first temperature or temperature-related variables.*", according to the reviewer's suggestion.

**Item 4:** 4. large spatial coherency. Reconstructions of the localized precipitation field or other variables related to precipitation are seldom (Cook et al., 2004; Cook et al., 2015b; Seftigen et al., 2015) because they require proxy records with more extensive distributions. In particular, the Palmer Drought Severity Index (PDSI) Atlases over the past millennium in North America

I think the authors do not mean more extensive spatial distribution, but rather a more dense proxy network that it would be the case for temperature.

**Response:** The sentence has been revised to "… *because they require more dense proxy network that it would be the case for temperature*", according to the reviewer's suggestion.

**Item 5:** 5. The climate field reconstruction method can be divided into the Empirical Orthogonal Function-based (EOF-based) method (Mann et al., 2009) and the point-to-point regression-based (PPR-based) method (Cook et al., 1999). The core function of the

I had real problems with this sentence. I think I understand what the authors mean, but the sentence can be really misleading. First, there are more 'families' of reconstruction methods - consider for instance the Bayesian Hierarchical Modelling Barcast, or the methods based on Canonical Correlation, or the more modern methods based on offline data assimilation (e.g. Steiger and Hakim) or even the method based on particle filters. Also, the RegEM method used by Mann et al is not really 'EOF-based. It is correct that Mann et al used an EOF pre-filtering within the RegEM method, but this is not required by the algorithm itself. Therefore, I do not think that this sentence is really correct. The authors may want to re-consider according with what I think they really want to say. They probably mean that statistical methods may include an EOF-prefiltering of the predictand or of the predictor or of both, or not pre-filtering at all. In the former case some small-scale information is lost - I think this is what the authors are pointing to.

**Response:** Following the reviewer's suggestion, we have rewritten the review of the reconstruction methods as proposed below:

[revised manuscript text omitted]

The 'left patterns' is unfortunate. It may be misinterpreted as 'left and right vectors' in SVD. I would rather used 'the discarded EOF patterns after EOF-truncation

**Response:** According to the reviewer's suggestion, these words have been changed with "*the discarded EOF patterns after EOF-truncation…*".

**Item 7:** 7. (Shi et al., 2015a) and the optimal information extraction (OIE) method (Yang et al., 2016). In theory, the PPR-based method maximizes the retention of spatial information, but this method requires a sufficient number of suitable proxy

I also had problems with the description of the OIE method, and also to figure out to what extent this method is different from the PPR method. This manuscript does not give enough details and refers to other previous manuscript by Shi et al. I have quickly looked into those papers and I cannot tell the difference between OIE and PPR. This may be my probable, or the problem in previous manuscripts, but I really would recommend to be much more specific here, and at least indicate the basic difference between OIE and PPR, and what are the advantages, if any, of OIE over PPR in this setting.

**Response:** Following the reviewer's suggestion, we have rewritten the review of these reconstruction methods to explain the difference between OIE and PPR. In our

interpretation, the foundation of the PPR method is that the climate field reconstruction should be obtained through a reconstruction for each grid point. The OIE method belongs to an indirect regression method group, which can be used to reconstruct a climate index or a climate field. The OIE method is based, as for the LOC method, on the principle that the reconstruction process should be based on the fact that the proxy records are functions of climate variables. The main differences between OIE method and other methods are that the correlation coefficients between the local instrumental climate data and the target climate data are used in the computation of the weights in the regression, and the regression coefficients are random variables with normal distribution and vary in the ranges between the classic linear regression and inverse regression to obtain an uncertainty estimation.

**Item 8:** 8 The precipitation (or the variable sensitive to precipitation) field reconstruction for a large-scale region using the PPR-based method is difficult when only one type of proxy records did not cover all reconstruction areas. For example, the tree-ring

This sentence is too cumbersome. I think I understand what it means, but the authors may consider rephrasing.

**Response:** The sentence has been revised to "*Selecting only one type of proxy records, with the associated limited spatial distribution, hinders the field reconstruction of precipitation (or of a variable sensitive to precipitation) for a large-scale region using the PPR-based method.*", according to the reviewer's suggestion.

**Item 9:** 9. regression and inverse regression. The LOC regression method has already been verified to efficiently retain low-frequency climate signals (Christiansen, 2011; Shi et al., 2012).

However, the LOC method has been shown to potentially overestimate the past variability. There is a comment and reply exchange on the Christiansen et al manuscript, and my interpretation of it is that Christian et al. also acknowledge that this could be a problem in certain circumstances.

**Response:** The sentence has been revised to "*The reconstructions based on the LOC method are assumed to better preserve the low-frequency climate signal compared to other methods, though they would overestimate the high-frequency signal (Christiansen and Ljungqvist, 2011).*", according to the reviewer's suggestion.

**Item 10:** 10 2.2 Tree-ring record

Please, be more specific here: three-ring width, isotopes, density, early wood density, etc.

**Response:** Following the reviewer's suggestion, we will add the metadata of all the proxy records in Table s1 in the revised manuscript.

**Item 11:** 11 To maximize the overlap lengths of the instrument data and proxy records, all treering records were extrapolated to AD 2000 using the RegEM algorithm

(Schneider, 2001). Here, the truncation parameters for the RegEM algorithm were set to

Extrapolation does not include new information and therefore it cannot increase the skill of the reconstructions. Was this step necessary for the OIE algorithm? if not, an explanation is required as to why the records were extrapolated.

**Response:** The extrapolation is necessary for the OIE method, because we used the correlation coefficient between the candidate proxy record and the reconstructed target to weight the candidate proxy records.

**Item 12:** 12 Discussion section. As I indicated in the preamble, this section actually contains further results, such as the superposed epoch analysis. It also contains the analysis of the link between the reconstructed precipitation and ENSO and the PDO. As it stands, it is a classical results section. The title' discussion' is misleading.

**Response:** Following the reviewer's suggestion, we have modified the structure of the paper and combined the results and discussion sections. We use thus the title 'Results and discussion'.

**Item 13:** 13. The superposed epoch analysis (SEA) between the precipitation, its PC1, and 35 large eruption events during AD 1470-1849 shows that volcanic activity as one important external forcing may affect the MJJAS precipitation anomalies variability for China (Fig. 8). Nevertheless, the signals are barely significant and there are similar averaged scores before and after the

These results are too cryptic. The SEA has not been mentioned before, so the reader is left wondering where this comes from: which eruptions have been included, how were they dated (the reconstructed volcanic forcing of Gao et al and of Crowley and Untermann does not always agree on the dating of the forcing maximum), how was the SEA itself conducted, for instance how many years prior to the eruptions were considered to define the pre-eruption mean, how was the statistical significance established, etc.

**Response:** Following the reviewer's suggestion, we have added the explanation of SEA method in the method section: "*The superposed epoch analysis (SEA) is traditionally used to analyse the influence of volcanic eruption on the climate, e.g., Bradley (1988). Here, the code to compute SEA has been downloaded from the website (http://blarquez.com/superposed-epoch-analysis-sea/). The period analysed (time window) are set as 20 years before and after each volcanic eruption event. The 90% confidence limit is estimated using the bootstrap procedure (Blarquez and Carcaillet, 2010). The eruption time series of Sigl et al. (2015) is used here because of the dating improvement compared to earlier estimates. Four categories of volcanic eruption events during the period from AD 1490 to AD 1829 are chosen following Zhuo et al. (2014)'s method which is based on the magnitude of their sulfate deposition in the Greenland ice-core records: (1) all Northern hemisphere eruption events (CNH0P)*

*according to Sigl et al. (2015), (2) CNH1/2P: the eruption events that have more than half, (3) equal (CNH1P), and (4) double (CNH2P) that of the sulfate deposition of the 1991 Mount Pinatubo eruption.".*

Moreover, we have updated Figure 8 as Figure s5 in the revised manuscript using the following Figure 1, which added the spatial patterns of the impact of the Northern Hemisphere volcanic eruption events on the precipitation field for the four categories of eruption (CNH0P, CNH1/2P, CNH1P, and CNH2P) of the volcanic events in Figure 1. *The SEA results applied to the mean precipitation anomalies (Fig. s5a), and its PC1 (Fig. s5b) during AD 1490-1829 shows that volcanic activity as one important external forcing might affect the MJJAS precipitation anomalies variability for China. Nevertheless, the signals are barely significant and there are similar averaged scores before and after the volcanic eruption year. Moreover, the spatial pattern of the impact of the Northern Hemisphere volcanic eruption events on the precipitation field (Figures s5c−f) is not consistent between the four categories of eruption (CNH0P, CNH1/2P, CNH1P, and CNH2P). This indicates that the response of MJJAS precipitation anomalies for China to Northern Hemispheric volcanic eruption is not robust.*

[Figure]

[Figure]

e) volcanic events with CNH1P  f) volcanic events with CNH2P

[Figure]

[Figure]

Figure 1 Superposed Epoch Analysis results applied to the mean precipitation anomalies (a), its PC1 (b), and the precipitation field (c-f) response to four categories (CNH0P, CNH1/2P, CNH1P, and CNH2P) of volcanic events as selected in Sigl et al. (2015) with 90% confidence limit during the period AD 1490-1829. The dashed lines in (a) and (b) are 90% confidence limit. The blank points in (c-f) identify statistically significant grid points at the 90% confidence level. The title of each panel in (c-f) indicates lag year from volcanic events.

**Item 14:** 14. Our results indicate thus that the south-north mode variability of precipitation anomalies in China carries very likely the fingerprint of ENSO evolution over the past 500 years, but the origin of the EOF1 and EOF3 patterns are not clearly established yet. This implies that the other factors such as North Atlantic Oscillation (NAO) (Wu et al., 2009; Zheng et al., 2016), interdecadal Pacific oscillation (IPO) (Song and Zhou, 2015), North Atlantic triple SST pattern (Ruan and Li, 2016) through the North Atlantic–Eurasia Teleconnection (AEAT) (Li et al., 2013a), the snow cover change of the Tibetan Plateau (Ding et al., 2009; Wu et al., 2012), and changes aerosol concentration (Li et al., 2016) may contribute to the reconstructed precipitation field modes during the pre-industrial period.

This conclusion is rather speculative. Why should EOF1 and EOF3 be related to the large-scale climate? they could be originated by regional processes in China.

**Response:** We agree with the reviewer's comment. This section is speculative but we wanted to include at this stage the reference to studies potentially useful to explain those patterns. Nevertheless, as we cannot explain the origins of EOF1 and EOF3 at the current stage, the regional processes in China is also a possible factor to affect them. Following the reviewer's suggestion, we have revised the part: "*Our results indicate thus that the south-north mode variability of precipitation anomalies in China carries very likely the fingerprint of ENSO evolution in tropical Pacific over the past 500 years. The origin of the EOF1 and EOF3 patterns is not clearly established yet, even though both of them maybe related to the western pacific subtropical high during the*

*instrumental period (Wu and Wang, 2002). Moreover, some studies show that other factors such as the North Atlantic Oscillation (NAO) (Wu et al., 2009; Zheng et al., 2016) and the North Atlantic triple SST pattern (Ruan and Li, 2016), the interdecadal Pacific oscillation (IPO) (Song and Zhou, 2015), the snow cover change of the Tibetan Plateau (Ding et al., 2009; Wu et al., 2012), and some regional processes in China may contribute to the precipitation field modes during the instrumental period. Thus, additional studies are then required to determine which of these processes might be related to EOF1 and EOF3 over the pre-industrial period.".*

**Item 15:** 15. Caption Figure 1. Please indicate what RSQ, RE, CE and uncertainty mean

**Response:** Following the reviewer's suggestion, we have added the full names of these terms in the caption of Figure 3, like as "*
[revised manuscript text omitted]

**II. Response to reviewer 2:**

**General comments:**

This is an interesting paper which introduces a newly generated annual warm season precipitation reconstruction over China. The reconstruction is based on the point-to-point regression-based method and a dense data network including 489 tree-ring width data, 2 tree-ring isotope data, 108 drought/flood index, and 1 long-term instrumental data. The verification results show good agreements between the reconstruction and instrumental data over eastern China. The paper is in itself interesting, but its structure and language needs to be improved. Since the methodologies have been commented by another reviewer, here I mainly add some comments about the proxy records.

**Response:** We would like to thank the reviewer for his/her constructive review, comments, and suggestions, which have helped us to greatly improve our manuscript. We have done our best to address the reviewer's concerns and modified the manuscript in light of the reviewer's suggestions, in particular to improve the structure and the language. Point-by-point responses to the reviewer's comments are listed below.

**Major comments:**

**Item 1:** 1. "Each record is required to be significantly correlated with one or more instrumental precipitation record at the 90% ($p < 0.1$) confidence level during the overlap period, based on both raw data and linearly detrended data."

How did you do the correlation analysis? How many nearby instrumental grid cells did you compare the proxy data with? It is surprising that all 489 tree-ring proxy records are "sensitive precipitation proxy records ".

**Response:** We calculated the Pearson's linear correlation coefficient between the targeted precipitation grid point and the candidate tree-ring record which is located in the range of the search radius, and select all tree-ring records that are positively and significantly related with one instrumental precipitation grid points during the period AD 1961-2000 at least ($r > 0$ and $p < 0.1$). Figure 1 shows the number of grid points that is reconstructed using each tree-ring record. e.g. a tree-ring record with "100" value means that it has significant relationship with 100 nearby instrumental precipitation grid points. This illustrates that all tree-ring records are significantly related to a precipitation grid point at least.

Thanks to your question, we have revised this statement, it is '*proxy records that can be significantly related to precipitation in the domain studied*' not '*sensitive precipitation proxy records*', because we do not determine whether the precipitation is the major limit factor of tree-ring width chronology and use all possible predictors to extend the spatial coverage.

[Figure]

Figure 1 The number of grid points which is reconstructed using each tree-ring record.

**Item 2:** 2. The quality of DWI before the instrumental period needs to be discussed.

**Response:** Following the reviewer's suggestion, we have added two Figures 2, 3 and a discussion on the quality of DWI in the revised manuscript, like as "*The DWI dataset is mainly derived from the local chronicles and started from 1470 (the sixth Year of Cheng hua reign in Ming Dynasty), which describe the onset, duration, areal extent, and severity of each drought or flood event in each province, China (Chinese Academy of Meteorological Science, 1981; Zhang, 1983). The experts, mainly from the provincial meteorological bureau provinces, China Meteorological Administration, Peking University, and Chinese Academy of Sciences, converted qualitative textual descriptions into quantitative data (Chinese Academy of Meteorological Science, 1981).*

*The reliability of DWI was described in previous studies (Zhang, 1983; Zhang, 1988), e.g., the homogeneity of DWI was demonstrated using the chi-square test (Zhang, 1983), and the reliability of DWI in spatial pattern was verified through comparison with the eigenvectors of the instrumental precipitation during the period AD 1951-1974 (Wang and Zhao, 1979).*

*However, DWI have still some weaknesses. The first one is time discontinuity (Zhang, 1983). Figure s1 is the time spans of 107 DWI records. This illustrates that most of DWI records are not continuous in time. The maximum (mean) number of missing values of 107 DWI records during the period AD 1470-2000 was 446 (157). Secondly, the DWI is unevenly distributed over space. Figure s2 shows that the location and number of DWI records during the period AD 1470-2000. On this figure, a value of "100" means that it has 100 observed values during the period AD 1470-2000. It indicates that 91 of 107 DWI records are located in eastern China (east of longitude 105°E), which is the economically developed region. There are only 16 DWI records west of longitude 100°E. Additionally, an uncertainty due to the subjective judgment is unavoidable for the*

*historical documentary, even though the different sources have been used to cross-validate the final reconstruction (Zhang, 1983; Man, 2009; Ge, 2011; Zheng et al., 2014a). Moreover, the range of values is within five grades and the DFI record is not an accurate precipitation value, thus it also limits the accuracy and ability to detect the extreme events (Zheng et al., 2014a).*

[Figure]

Figure 2 The time span of each Dryness/Wetness index.

[Figure]

Figure 3 The number of observation of each Dryness/Wetness index during the period AD 1470-2000.

*In order to improve the quality of the DWI dataset, Professor Zhang De'er lead a team of scientists, that carried out research during 20 years to identify the weather events in China over the past 3000 years. This resulted in the publication of a book, entitled "A compendium of Chinese meteorological records of the last 3,000 years". Each record has been carefully cross-checked from more than 8,000 historical documents (Zhang, 2004). However, the updated dataset is not archived in publish repository so far."*

**Item 3:** 3. As you mentioned the precipitation/ PDSI reconstructions over East Asia from Cook et al. (2010) and Feng et al. (2013), have you compared this reconstruction with theirs?

**Response:** Following the reviewer's suggestion, we have calculated the correlation between Cook's PDSI reconstruction (Cook et al., 2010) with our precipitation reconstruction in Figure 4.

Figure 4 shows that the spatial correlation between them. We have included Figure 4 as supplementary material and discussed it in the revised manuscript, like as "*Strong correlations appear in the northeastern Tibetan Plateau, where there are longest and most abundant tree-ring width chronologies in China. Here, the precipitation is possibly a primary control factor of the tree-ring width chronology (Zhang et al., 2003c; Yang et al., 2014b). There are weak correlations between the reconstructions in eastern China even through the correlation coefficients are significant in some regions. The MADA is not consistent with the DWI records in eastern China, since only very few and short tree-ring width chronologies in eastern China are used to reconstruct the MADA (Yang et al., 2013a; Kang et al., 2014; Yang et al., 2014a; Zheng et al., 2014b; Ge et al., 2016).*"

[Figure]

Figure 4 Spatial correlation between the PDSI reconstruction (Cook et al., 2010) and the precipitation field reconstruction obtained in this study. The dark spots mean the significant levels of the correlation coefficients at the 99% confidence level.

It is a pity that the precipitation reconstruction (Feng et al., 2013) is not achieved in any published database. However, we have added Figure 5 as Figure 6 in the revised manuscript for qualitative comparison of five historical Chinese droughts.

*The selection of five drought periods follows Feng et al. (2013) but the spatial patterns in our reconstruction display some clear differences from their results. A "north drought with south flooding" dipole pattern in eastern China is seen in Figures 6a−d, and there is a triple pattern in eastern China in Fig. 6e. A similar dipole pattern can be found in Figures 5b and 5d (Feng et al., 2013) for two drought events (AD 1586−89 and AD 1876−78) but during four out of five drought periods, most of northeastern China is relatively humid situation in our study, which is not consistent with Feng et al. (2013)'s Fig. 5. An exceptional drought condition in northeastern China appears in the 1876-1878 drought event, which is the most severe drought of five events in eastern China.*

[Figure]

Figure 5 Spatial patterns of the May–September precipitation field relative to the 1961–90 climatological mean during five severe droughts in China.

Moreover, the reconstruction skills are low in semiarid and arid regions (Feng et al., 2013), because there are very few tree-ring chronologies in Western China in that reconstruction. We used some tree-ring width chronologies in Western China to compensate this weakness.

**Item 4:** 4. "A total of 242 of 491 tree-ring chronologies were extrapolated. The maximum and mean extrapolation lengths of the 242 chronologies were 24 years and

As you mentioned "all tree-ring records were extrapolated to AD 2000", that means lots of infilling data are between 1981-2000CE, which overlaps with the calibration period (1981-2000CE). Have you considered using a longer calibration period and leave out the last ten years?

**Response:** The extrapolation is necessary for the OIE method, because we used the correlation coefficient between the candidate proxy record and the reconstructed target to weight the candidate proxy records. Following the reviewer's suggestion, we have used a longer calibration period (1961-1990) and the last ten years as the verification period (1991-2000). The corresponding contents have revised according to the updated results.

**Item 5:** The second and third leading EOF patterns, the south-north dipole and the sandwich triple pattern, are indeed found in instrumental period. And they are closely related to the movement and intensity of the western pacific subtropical high (WPSH). The sea surface temperature can influence the rainfall through WPSH, but correlation maps between ENSO and summer precipitation are often noisy (Wu and Wang 2002). So, why would the signal of ENSO in rainfall be so strong in the reconstructions if it is noisy in instrumental rainfall data?

**Response:** This is indeed an interesting point. We have calculated the running correlation between the 17 reconstructed ENSO indices and the precipitation anomalies averaged over Yangtze River region and Huai River region in Figures 6a-b. We have followed Wu and Wang (2002) for the region definition. All data were filtered to obtain the interannual components when calculating running correlation coefficients using the Ensemble Empirical mode decomposition (EEMD) method, because the ENSO variability is in the interannual 3–7 year period band (Trenberth, 1997). Results indicate that the relationship between ENSO and summer precipitation in China are not stable in the reconstructions, which is consistent with the instrumental period (Wu and Wang, 2002). However, we have calculated the correlation coefficient between the 17 ENSO indices and the precipitation for the full period, and all show that small and positive correlation. Thus, a more robust link between ENSO and summer precipitation in China may be found, if we analyze the full period. Moreover, the explanatory variance of the second dominant mode of precipitation is only about 11.2%. This indicates that different factors may substantially affect the spatial pattern of the precipitation field, e.g. the snow cover change of the Tibetan Plateau (Ding et al., 2009).

Finally, we have added the statement that the two leading EOF patterns are closely related to the western pacific subtropical high (Wu and Wang, 2002) in the revised manuscript, like as "*The origin of the EOF1 and EOF3 patterns over the pre-industrial period is not clearly established yet, even though both of them maybe related to the movement and intensity of the western pacific subtropical high during the instrumental period (Wu and Wang, 2002).*".

[Figure]

Figure 6 The 101-year running correlation coefficients of the interannual components of 17 ENSO indices with the reconstructed precipitation anomalies averaged over Yangtze River region (28°−32°N, 110°−120°E; a) and Huai River region (32.5°−35.5°N, 115°−120°E; b). The interannual components were obtained using the Ensemble Empirical mode decomposition (EEMD) method.

**Item 6:** 6. In Figure 9, which method did you use for the correlation analysis? How does it look using winter temperature in the Nino 3.4 region?

**Response:** In Figure 9, the Pearson's sample linear cross-correlations at lag 0 is used for the correlation analysis, and the effective number of degrees of freedom is calculated following Zhao et al. (2016) to access the significance of the correlation. The unified ENSO proxy (UEP) (McGregor et al., 2010) is used to calculate the relationship between the precipitation field and the ENSO index. This index is the first PC of ten reconstructed ENSO indices, and is scaled to the instrumental HadSST1 annual mean (July–June) Niño 3.4 region sea surface temperature anomalies (McGregor et al., 2010). We have added this explanation in the method section. Moreover, we have calculated 
[revised manuscript text omitted]

---

## Author Response (AR2)

**Interactive comment on "Multi-proxy reconstructions of May–September precipitation field in China over the past 500 years"**

Feng Shi[1,2*], Sen Zhao[3,4], Zhengtang Guo[1,5,6], Hugues Goosse[2], Qiuzhen Yin[2]

[1]Key Laboratory of Cenozoic Geology and Environment, Institute of Geology and Geophysics, Chinese Academy of Sciences, Beijing, 100029, China
[2]Georges Lemaître Centre for Earth and Climate Research, Earth and Life Institute, Université catholique de Louvain, Louvain-la-Neuve, 1348, Belgium
[3]Key Laboratory of Meteorological Disaster of Ministry of Education, and College of Atmospheric Science, Nanjing University of Information Science and Technology, Nanjing, 210044, China
[4]School of Ocean and Earth Sciences and Technology, University of Hawaii at Mānoa, Honolulu, HI, 96822, USA
[5]CAS Center for Excellence in Tibetan Plateau Earth Sciences, Beijing, 100101, China
[6]University of Chinese Academy of Sciences, Beijing, 100049, China

*Correspondence to*: Feng SHI (shifeng@mail.iggcas.ac.cn)

- In blue: referees' comments

- In black: our answers

- In black italic: what we will propose to add in the text

————————————————————————————————————————————————

**General comments:**

The two reviewers are both satisfied with the revised version of your manuscript. However, there is still some clarification needed regarding the data availability. Please pay close attention to the following requirements when revising the manuscript:

**Major comments:**

**Item 1:** 1) Expand the data availability section to explain that: Citations to the persistent identifiers for the original data used in this study are listed in Table S1.

**Response:** Following the team's suggestion, we have expanded the data availability section to explain the identifiers, as following '*A link (URL) to the original record is included in Table S1, which corresponds to the location where the original record is stored in a public repository.*'

**Item 2:** 2) Submit the primary outcome of the data analyses to a public repository and include the Data Citation. This includes (a) the detrended and infilled version of all tree-ring

chronologies, (b) the resulting precipitation time series reconstruction with and without 9-year smoothing, (c) the IMFs of the reconstruction time series at multiple temporal scales (Figs 5 and S1), and (d) the reconstructed precipitation value for each grid point at a reasonable time resolution, possibly decadal.

**Response:** Following the team's suggestion, we have submitted the metadata with the data citation in Table S1 and will submit the outcome of the data analyses to the NOAA website. Specifically, (a) We will submit the detrended and infilled version of the 362 tree-ring chronologies obtained to the NOAA website. The other 10 tree-ring width chronologies have already been detrended, and can be obtained from the website of the Chinese Meteorological Data Service Center (CMDC). The script of the Regularized Expectation Maximization (RegEM) method is available at http://climate-dynamics.org/software/#regem and can be easily used to extend these chronologies to AD 2000. (b) We will submit the resulting precipitation time series of reconstruction and climate model simulations with and without 9-year smoothing in Fig.7 to the NOAA website. (c) We will submit the IMFs of the reconstruction time series at multiple temporal scales (Figs 5 and S1) to the NOAA website. (d) The gridded precipitation data will be submitted at an annual time resolution in the NOAA website to allow analyzing extreme events, and it can be filtered to any time resolution according to the motivation, e.g. decadal time scale.

**Item 3:** 3) Submit Table S1 as part of the data archive for this study

**Response:** Following the team's suggestion, we will submit Table S1 as part of the data archive for this study.

**Item 4:** 4) Double check that all of the "original data URLs" are correct. Note: https://www.ncdc.noaa.gov/paleo/study/471012693 is returns an error.

**Response:** Following the team's suggestion, we have double checked all of the "original data URLs". The URL (https://www.ncdc.noaa.gov/paleo/study/471012693) is corrected with https://www.ncdc.noaa.gov/paleo/study/12693.

**Item 5:** 5) All of the URL links to the CMDC seem to land on the sample page. I suspect that it's because I'm not a registered user. Please add an explanation to the data availability section to alert users that registration is required or whatever is needed to lower the barrier to finding the data.

**Response:** Yes, you are right. The registration is necessary to these records. Follow the team's suggestion, we have added an explanation in data availability section, as following '*The registration is required to obtain these records from CMDC.*'

**Item 6:** Finally, the language needs some improvement to make it more readable. I suggest having a native English speaker looking over the manuscript.

**Response:** We will do our best to improve the language.